# Mechano-gated iontronic piezomemristor for temporal-tactile neuromorphic plasticity

Xiao Wei[1,2,3,7], Zhixin Wu [1,2,7], Hanfei Gao[3,7], Shiqi Cao[4,7], Xue Meng[1,2], Yuqun Lan[5], Huixue Su[1,2], Zhenglian Qin[1,2], Hang Liu[1,2], Wenxin Du [6], Yuchen Wu [1,2,3] ✉, Mingjie Liu [6] ✉ & Ziguang Zhao [1,2] ✉

In bioneuronal systems, the synergistic interaction between mechanosensitive piezo channels and neuronal synapses can convert and transmit pressure signals into complex temporal plastic pulses with excitatory and inhibitory features. However, existing artificial tactile neuromorphic systems struggle to replicate the elaborate temporal plasticity observed between excitatory and inhibitory features in biological systems, which is critical for the biomimetic processing and memorizing of tactile information. Here we demonstrate a mechano-gated iontronic piezomemristor with programmable temporal-tactile plasticity. This system utilizes a bicontinuous phase-transition hetero-gel as a stiffness-governed iontronic mechanogate to achieve bidirectional piezoresistive signals, resulting in wide-span dynamic tactile sensing. By micro-integrating the mechanogate with an oscillatory iontronic memristor, it exhibits stiffness-induced bipolarized excitatory and inhibitory neuromorphics, thereby enabling the activation of temporal-tactile memory and learning functions (e.g., Bienenstock–Cooper–Munro and Hebbian learning rules). Owing to dynamic covalent bond network and iontronic features, reconfigurable tactile plasticity can be achieved. Importantly, bridging to bioneuronal interfaces, these systems possess the capacity to construct a biohybrid perception-actuation circuit. We anticipate that such temporal plastic piezo-memristor devices for abiotic-biotic interfaces can serve as promising hardware systems for interfacing dynamic tactile behaviors into diverse neuromodulations.

Tactile perception is constructed by a sophisticated neural transmission process through the synergistic interaction between mechanosensitive piezo channels and neuronal synapses[1] (Fig. 1a). Piezo ion channels, which are sensitive to mechanical forces, can capture and detect various stress stimulations, activating ion flux transmission to form action potentials[2,3]. Neuronal synapses further generate excitatory or inhibitory signals featuring spike-timing dependent plasticity (STDP), which is proposed to underlie the formation of memory traces, dynamic tactile perception, and cognitive behaviors[4–6].

---

[1]School of Future Technology, University of Chinese Academy of Sciences, 100190 Beijing, PR China. [2]Key Laboratory of Bio-Inspired Materials and Interfacial Science, Technical Institute of Physics and Chemistry, Chinese Academy of Sciences, 100190 Beijing, PR China. [3]Suzhou Institute for Advanced Research, University of Science and Technology of China, Suzhou 215123 Jiangsu, PR China. [4]Orthopaedics of TCM Senior Department, The Sixth Medical Center of Chinese PLA General Hospital, 100048 Beijing, PR China. [5]State Key Laboratory of Nonlinear Mechanics, Institute of Mechanics, Chinese Academy of Sciences, 100190 Beijing, PR China. [6]School of Mechanical Engineering and Automation, Beihang University, 100191 Beijing, PR China. [7]These authors contributed equally: Xiao Wei, Zhixin Wu, Hanfei Gao, Shiqi Cao. ✉e-mail: wuyuchen@iccas.ac.cn; liumj@buaa.edu.cn; zhaoziguang@ucas.ac.cn

Conventional tactile sensors, particularly those based on conductive polymers, gel materials, and nanocomposites, transduce stress signals into piezoresistive, capacitive, or piezoelectric outputs, which primarily reflect unidirectional stress intensity variations[7–11]. Recently, with the integration of neuromorphic memristor units, artificial tactile systems have been developed to realize tactile perception with synaptic-like functions[12,13], which have attracted significant attention for their promising applications in biomimetic machines[14–16], neuroprosthetics[17,18], and neuromorphic computing[19,20]. However, compared to the dynamic tactile processing in bioneuronal systems, most existing tactile systems only possess monotonic neuromorphic features. In response to varying tactile behaviors, these systems, with their limited sensing capabilities, struggle to provide elaborate temporal plasticity involving excitatory and inhibitory neuromorphic signals. This constraint hinders the development of complex tactile processing and memory functions, such as tactile memory erasure and reversible modulation. Achieving nuanced temporal-tactile neuromorphic plasticity within an artificial perception system remains a challenge.

Here, we report a mechano-gated iontronic piezomemristor (MIPM) to achieve programmable temporal-tactile neuromorphic plasticity (Fig. 1b). This system utilizes a bicontinuous heterogel as a stiffness-governed iontronic mechanogate. In contrast to existing gel-based sensors that offer unidirectional tactile perception[21,22], the phase-transition-induced switchable stiffness of our heterogel introduces distinctive mechano-gated properties. These properties facilitate the transduction of complex stress information into bidirectional (both negative and positive) piezoresistive signals within a biologically relevant temperature range of 25 °C to 40 °C (Fig. 1c). Owing to the micro-integrated coordination between iontronic mechanogate and oscillatory memristor in MIPM, the bipolarized excitatory and inhibitory neuromorphic signals with STDP features can be generated, further activating high-order temporal-tactile memory and learning functions (e.g., Bienenstock–Cooper–Munro (BCM) and Hebbian learning rules). Such MIPM systems, with their dynamic covalent bond network and iontronic features, also offer reconfigurable capabilities for tactile neuromorphic plasticity. Within wide-span dynamic tactile sensing with sensitivities ranging from −0.112 to 11.92 kPa⁻¹, the related neuromorphic plasticity can be controlled. Meanwhile, utilizing multi-species ions as signal carriers, MIPM system exhibits various iontronic features. In this study, the MIPM demonstrates the capability to construct a biohybrid perception-actuation circuit for precisely modulating biological neural interfaces, thereby enabling various levels of the limb movement control in response to complex stress stimulations. Such piezomemristor within abiotic-biotic systems can serve as a promising hardware implementation to interface dynamic tactile perceptions with diverse neuromodulations.

## Results

### Switchable stiffness iontronic mechanogate

The intrinsic stiffness factor of biological mechanoreceptors can affect the sensitivity of mechanosensitive piezo channels, thereby enabling the differential transduction of touch signals[2]. Mimicking such mechanism within biological mechanoreceptors, we introduce the bicontinuous iontronic heterogel as a switchable stiffness iontronic mechanogate capable of transducing pressure signal into bidirectional piezoresistive signals. An orthogonal polymerization-induced phase separation strategy was utilized to fabricate the bicontinuous heterogels, which synergistically integrated both a stiff vitrimer and a soft ion-liquid gel (ILgel) as distinct continuous phase frameworks (Fig. 2a, b and Supplementary Fig. 1). Nano computed tomography (Nano CT) scanning and atomic force microscopy (AFM) measurements confirm that the vitrimer and the ILgel domains are seamlessly interpenetrated, with each phase domain forming a continuous

structure throughout the heterogel (Supplementary Fig. 2). Derived from the DSC results, the bicontinuous heterogels exhibit a significant endothermic peak around 37.5 °C, indicating a distinct thermal-induced phase transition of the vitrimer phase at this temperature (Supplementary Fig. 3b). In these bicontinuous heterogels, the ILgel phase serves as ion transport channels and the vitrimer phase manifests the phase-transition-induced switchable stiffness that can influence ion transmission behavior in a piezoresistive process. Three-dimensional finite element analysis (FEA) demonstrates the correlation between the bicontinuous structures and iontronic piezoresistive features (Fig. 2c, d). In the stiff state, the high modulus of vitrimer phase restricts lateral deformation of ILgel phase, where axial compression primarily shortens the ion transmission channels under compressive strain ($\varepsilon = 50\%$). This compression slightly increases the effective cross-sectional area of the ILgel phase with ion channels from 49.4 to 49.9%, resulting in negative piezoresistivity sensing behavior ($\Delta R < 0$). When the heterogel is switched to a soft state due to phase transition, the significantly reduced modulus of vitrimer allows for substantial lateral expansion of heterogel materials under compression. This lateral deformation causes the ILgel phase to become partially interrupted and disconnected, resulting in a substantial decrease in the effective cross-sectional area from 49.4% to 31.2%, leading to positive piezoresistive features ($\Delta R > 0$).

The switchable stiffness features and bidirectional piezoresistivity properties of bicontinuous heterogels were further evaluated experimentally. Within biologically relevant temperature range of 25 °C to 40 °C, the mechanical modulus of the bicontinuous heterogel varies by three orders of magnitude (Supplementary Fig. 3). Infrared thermography results in Supplementary Fig. 4 show the rapid thermal response of the heterogels, which enables efficient transitions between stiff states and soft states. Rheological measurements reflect the stable switchable stiffness feature from 9 kPa to 13 MPa (Supplementary Fig. 5). Figure 2e and Supplementary Fig. 6 demonstrate the stiffness-induced piezoresistivity features of the heterogel mechanogate. In their stiff state, the heterogels exhibit highly sensitive negative piezoresistive properties ($S = 4.34$ kPa⁻¹). Conversely, in their softened state, they show positive piezoresistivity with a sensitivity of −0.034 kPa⁻¹. This stiffness-induced bidirectional piezoresistivity features exhibits excellent stability, remaining virtually unchanged after 100 cycles of soft/stiff state switching (Supplementary Fig. 6). Upon increasing the step-up pressure from 0 kPa to 50 kPa and subsequently returning it to the initial state, the heterogel mechanogate exhibits excellent segment stability and clear bidirectional piezoresistivity properties (Fig. 2f). Meanwhile, Fig. 2g displays reliable negative and positive piezoresistive signal responses under loading and unloading cycles with different pressures. Distinguished from existing typical unidirectional piezoresistive systems (Supplementary Table 1), including ion hydrogels, ILgels, nanocomposites, conductive polymers, aerogel and liquid metals, our bicontinuous heterogels exhibit bidirectional stiffness-gated piezoresistivity. This material with bicontinuous structures integrates both positive and negative piezoresistive properties, combining diverse sensitivity with excellent stability, effectively enhancing sophisticated sensory capabilities for tactile memristor systems.

### Temporal-tactile plasticity based on excitatory and inhibitory neuromorphics

During the tactile sensing process in bioneuronal systems, the synergistic interaction between mechanosensitive piezo channels and neuronal synapses generates excitatory and inhibitory signals, creating cumulative temporal effects within neural networks. The temporal plasticity, which depends on the relative timing of these excitatory and inhibitory signals, is crucial for the regulation of neural circuits. This process endows the circuits with complex perception, memory, and dynamic learning functions[5,6].

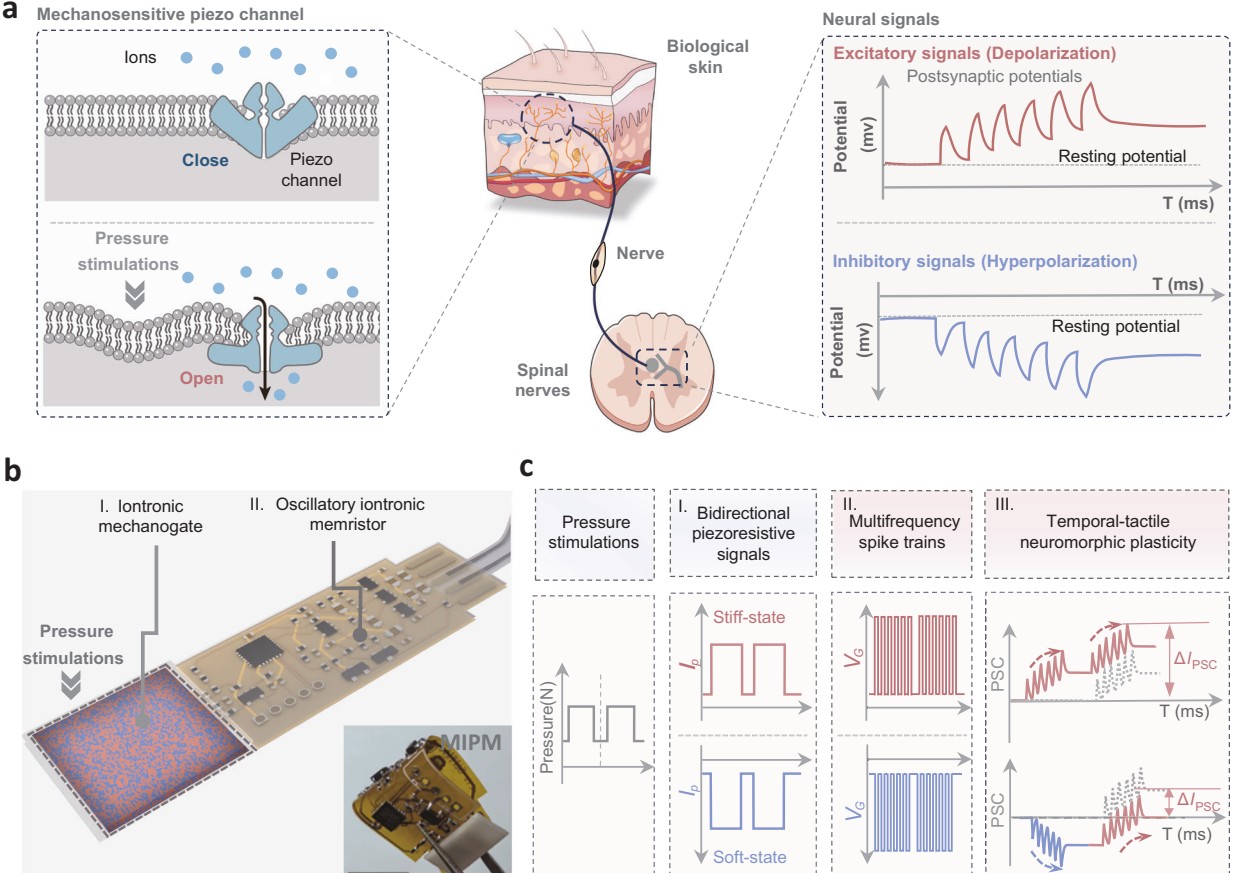

**Fig. 1 | Mechano-gated iontronic piezomemristor (MIPM). a** Schematic of the tactile perception mechanism in bioneuronal systems. The synergistic interaction between mechanosensitive piezo channels and synapses of spinal nerves efficiently converts tactile signals into complex neural signals with excitatory and inhibitory features. **b** Concept and structure of the MIPM. The MIPM consists of a switchable stiffness iontronic mechanogate, and an oscillatory iontronic memristor. The scale bar is 1 cm. **c** Schematic diagram of tactile signal conversion and transmission in the MIPM system. Complex pressure information is captured by the switchable stiffness mechanogate, forming bidirectional piezoresistive signals. These signals are subsequently converted into positive and negative spike trains, then processed and outputted as neuromorphic excitatory or inhibitory signals. In this MIPM system, the temporal tactile plasticity can be realized through the integration of bipolarized neuromorphic signals.

We developed a MIPM system featuring biomimetic temporal-tactile plasticity based on excitatory and inhibitory neuromorphic signals. The structure and design of the MIPM, incorporating a mechanogate and an oscillatory iontronic memristor, are illustrated in Fig. 3a and Supplementary Figs. 7 and 8. For the iontronic memristor within the MIPM, the mixed ionic organic semiconductor poly(3,4-ethylenedioxythiophene) (PEDOT) doped with poly(styrene sulfonate) (PSS) is assembled to micro-wires array in-situ using a confined assembly strategy, connecting with the oscillating digital circuit through an ionic conductive gel (Supplementary Figs. 9 and 10). The ordered array structures endow the memristor with exceptional stability, exhibiting highly reproducible hysteresis characteristics over 100 consecutive bidirectional scanning cycles (Supplementary Fig. 11). Detailed fabrication processes and the structural features of the oscillatory iontronic memristor are provided in Supplementary Information. In MIPM systems, a switchable-stiffness heterogel mechanogate captures and recognizes pressure stimuli, generating bidirectional piezoresistive signals. The oscillator module with an integrated microcontroller encodes these piezoresistive signals into positive and negative spike trains based on their direction and magnitude. This frequency-modulated spike trains encoding preserves the temporal and intensity information of the original piezoresistive signals, while normalizing spike amplitudes (Supplementary Figs. 12 and 13). Under the driving of these frequency-encoded bipolar sequences, ion injection and removal processes regulate the characteristics of the PEDOT:PSS iontronic memristor. Specifically, the coupling transmission of ions and electrons within PEDOT:PSS can efficiently control the drain current ($I_D$)[23,24], thereby enabling the generation of tunable neuromorphic excitatory/inhibitory signals (Supplementary Fig. 10). Furthermore, as shown in Supplementary Fig. 14, the temporal patterns of positive and negative spike trains influence the synaptic potentiation behaviors of the memristor, which in turn modulate the device output (postsynaptic current, PSC). This mechanism enables the device to realize excitatory and inhibitory neuromorphic signal outputs with STDP.

The excitatory and inhibitory neuromorphic properties of the MIPM were further investigated in response to complex pressure stimulations. Different pressure stimulations were used as touch signal inputs to the stiff-state mechanogate of the MIPM. The increase in pressures produces concomitant negative spike trains of higher frequency, resulting in large changes in the amplitudes of PSCs, that is, a ~40% output difference between 10 and 20 kPa inputs and about a ~70% output difference when extending to 50 kPa (Fig. 3b and Supplementary Fig. 15). Conversely, in the soft state, higher frequency positive spike trains transduced from larger piezoresistive signals result in a more pronounced reduction of PSC, thereby exerting a considerable inhibitory effect. The increased pressures in both stiff

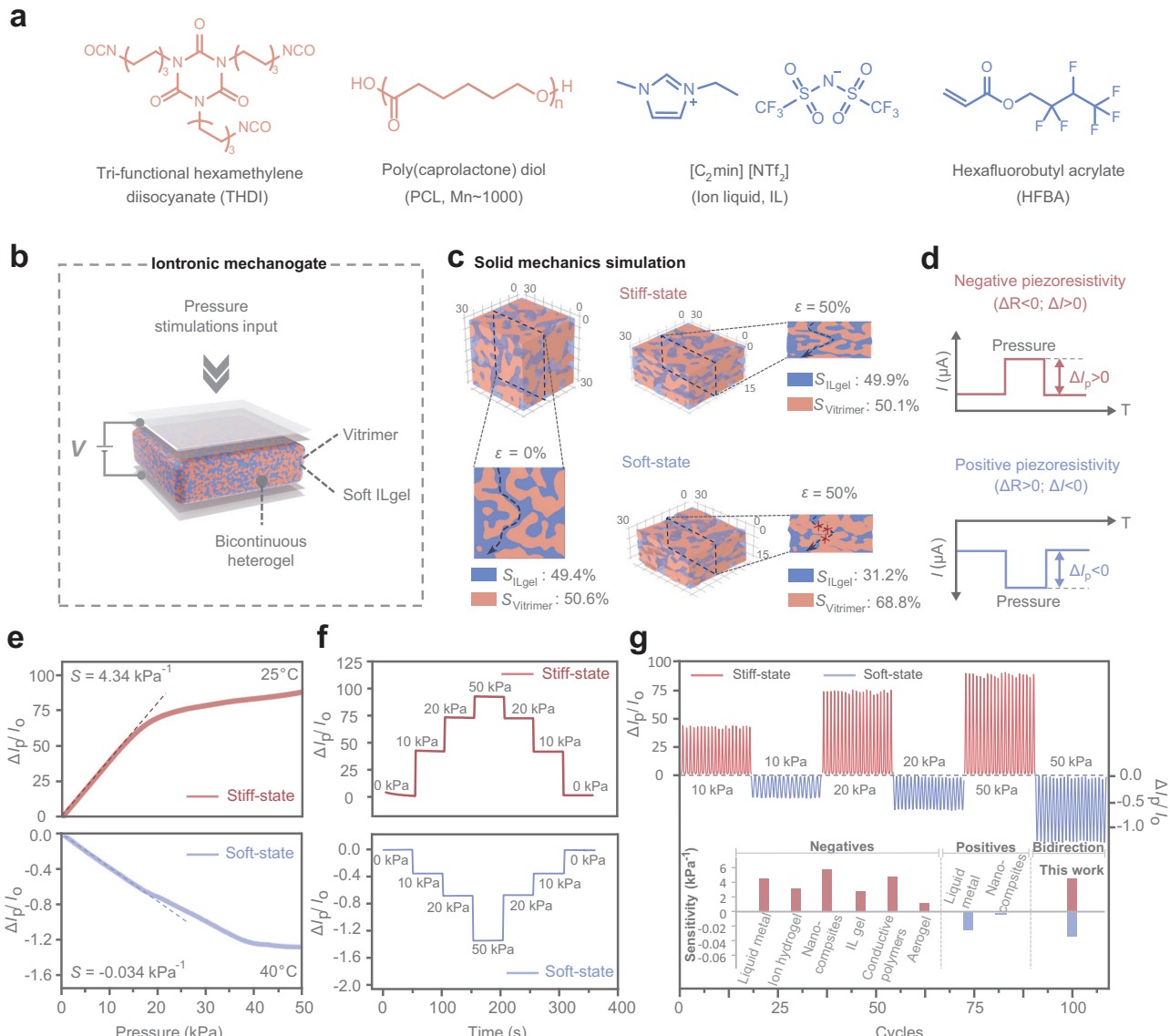

**Fig. 2 | Switchable stiffness iontronic mechanogate. a** Chemical structures of reacted precursor components within the bicontinuous heterogel. **b** Structural features of the stiffness iontronic mechanogate based on a bicontinuous heterogel. The vitrimer (orange) and ILgel (blue) domains formed bicontinuous structure through mutual interpenetration. **c** Three-dimensional finite element analysis of the bicontinuous heterogels with distinct stiffness states. The bicontinuous heterogel exhibits distinct strain-induced structural variations in stiff and soft states under compression ($\varepsilon = 50\%$), where the effective cross-sectional area of the ILgel phase ($S_{\text{ILgel}}$) and Vitrimer phase ($S_{\text{Vitrimer}}$) undergo opposite changes, leading to bidirectional piezoresistive behaviors. **d** The current signal ($I$) of the heterogel mechanogate. The negative and positive piezoresistivity characteristics correspond to the stiff and soft states, respectively. **e** The stiffness-induced

piezoresistivity ($\Delta I_{\text{p}}/\Delta I_{\text{o}}$, where $\Delta I_{\text{p}}$ and $\Delta I_{\text{o}}$ corresponding to the current of mechanogate under loading and unloading pressure, respectively) features of the iontronic mechanogate, demonstrating the negative and positive sensitivity ($S$). **f** The segment stability test of bidirectional piezoresistivity under the step-up pressure from 0 kPa to 50 kPa. **g** Negative and positive piezoresistive signal responses of the bicontinuous heterogels under cycle loading (loading pressure: 10 kPa, 20 kPa, 50 kPa). The inset shows the statistics of piezoresistivity features in existing typical pressure sensors (including ion hydrogel, ILgel, nanocomposite, conductive polymer, aerogel and liquid metal) and the heterogel. Distinguished from their unidirectional piezoresistive features, our mechanogate demonstrates a bidirectional stiffness-gated piezoresistivity.

and soft states further induce a change in the device's current decay process from volatility to non-volatility, analogous to the transition from short-term plasticity (STP) to long-term plasticity (LTP) in biological synapses (Fig. 3c, d). To better describe the decay current, curves are used to fit the relaxation process after pressure removal. With the increase of pressure, the current descent curves can be fitted well using a double exponential decay equation, as follows: $I(t) = C_1\exp(-t/\tau_1) + C_2\exp(-t/\tau_2) + C_0$, where $C_0$, $C_1$, and $C_2$ are fitting coefficients, and $\tau_1$ and $\tau_2$ are the characteristic time constants[25]. The values of $\tau_1$ and $\tau_2$ indicate the coexistence of the rapid and slow relaxation in the

descent process under high-intensity pressure. The rapid and slow decays of PSC are associated with the ion transport and diffusion kinetics within the PEDOT:PSS. Ion transports are significantly slower compared to the motion of electrons and holes, and this lagging effect leads to distinct time-scale current variations and memory effects in the iontronic memristor[26]. Under low-pressure stimulations, low-frequency spike trains induce limited ion accumulation in the PEDOT:PSS, where the coupling of ions and electrons results in transient conductivity changes. After spike trains are removed, these ions rapidly diffuse back, leading to the spontaneous recovery of the

memristor conductivity, which corresponds to the short-term memory characteristics. As the pressure increases, high-frequency electrical spike trains drive substantial ion accumulation within the PEDOT:PSS.

The enhanced coupling of abundant accumulated ions and electrons in PEDOT:PSS involves conformational changes of the PEDOT chains and rearrangement of the PSS chains, leading to a more stable ionic storage

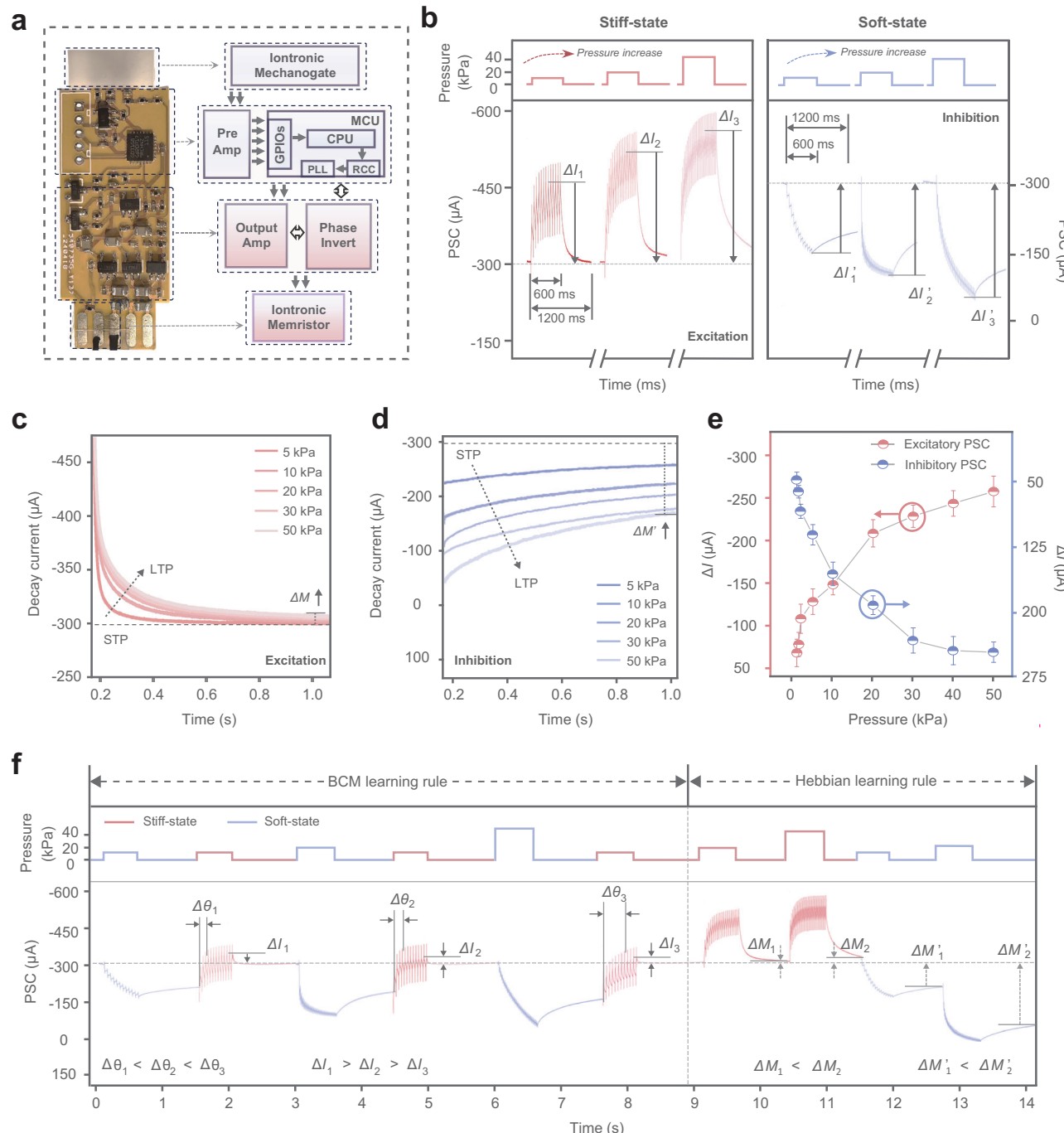

**Fig. 3 | Temporal-tactile plasticity based on excitatory and inhibitory neuromorphics. a** Design of the MIPM. The schematic shows the processing pathway of tactile signals. **b** The postsynaptic current (PSC) curves corresponding to 10 kPa, 20 kPa, 50 kPa loading pressure. As the loading pressure increased, the absolute value of the change in postsynaptic current amplitude ($\Delta I$) enhanced. **c, d** The decay curves of MIPMs' PSCs in stiff-state and soft-state, demonstrating transition from short-term plasticity (STP) to long-term plasticity (LTP) of MIPM. **e** The absolute value of the change in PSC amplitude as a function of pressures applied to MIPMs in stiff and soft state. Data points represent average values from ten measurements. The bipolarized plasticity facilitates high-precision recognition of complex tactile signals. **f** Temporal-tactile plasticity of MIPM with two heterogel mechanogates at different stiffness states demonstrating BCM learning rule and

Hebbian learning rule. BCM learning: Three controlled trials (different pressures of 10 kPa, 20 kPa and 50 kPa, respectively) as the experienced historical activities activated three non-volatile current levels from the resting current. Subsequent testing pressure (10 kPa) input yields the spike trains with the same frequency, but achieving different synaptic responses ($\Delta I$ and $\Delta\theta$, where $\Delta\theta$ represents the time interval between the neuromorphic signal generation and its reaching to the resting current level). This result illustrates MIPM's BCM learning function. Hebbian learning: MIPM modulated the postsynaptic current variation of subsequent pressure stimulations by controlling excitatory and inhibitory non-volatile current levels ($\Delta M_1$ and $\Delta M_1'$, where $\Delta M$ represents the absolute value of the change in non-volatile current) based on different historical pressures, demonstrating Hebbian learning rules.

state. This gives rise to the non-volatile current with a retention time extending to the minute scale ($\Delta M$ and $\Delta M'$), corresponding to the long-term memory characteristics. This result indicates that the switching between volatile and non-volatile memory states can be achieved by controlling pressure stimulations. Notably, due to the differences in the coupling effect of electrons and different ions within the PEDOT:PSS, asymmetric non-volatile excitatory and inhibitory currents are observed relative to the pristine state, with higher shifts for non-volatile inhibitory currents. Figure 3e demonstrates the modulation effect of the applied pressure on the excitatory and inhibitory PSC, and this bipolarized plasticity facilitates high-precision recognition of complex touch signals. Additionally, Supplementary Fig. 16 further exhibits the piezoresistive behavior and corresponding postsynaptic currents of the MIPM system across a wide temperature range, demonstrating the similar PSC modulation effect of the applied pressure.

The synaptic weight, or conductance, can be modified during learning processes on both short and long timescales[5,6]. Similarly, our MIPM system features multi-timescale plasticity in its bipolarized neuromorphic signals, enabling reversible modifications of the memristor's conductance. Thus, these devices can emulate basic learning rules observed in biological synapses. For instance, the BCM rule suggests that synaptic weight modulation is frequency-dependent and influenced by the history of neural activities[27,28] In the MIPM system, the BCM rule is realized through programmed temporal pressure stimulations (Fig. 3f and Supplementary Fig. 17). To eliminate the potential influence of temporal inhomogeneity switching on memory and learning effects, two heterogel mechanogates featured different stiffness states were used alternately within the MIPM system. We conducted three controlled trials of gradient-increased pressure on the soft-state mechanogate as history activities, followed by three constant pressures on the stiff-state mechanogate as testing trials. The three sets of spike trains (successive positive spikes) at gradient-increased frequencies, corresponding to the three sets of history pressure, resulted in increasing inhibitory plasticity and adjusted the MIPM's the long-term memory to different levels below the resting current. The subsequent positive spike trains corresponding to the stiff-state pressure, despite having the same frequency, caused different synaptic responses ($\Delta I$, $\Delta \theta$) due to the long-term memory of the previous spike trains, which demonstrates that the device's synaptic plasticity is history-dependent. Notably, adding a small positive spike train after each negative spike train can not only be used to explore $\Delta I$ but also rapidly decay the PSC to the resting level due to its short-term plasticity. This reset operation allows the plasticity of the MIPM to be tuned, adjusting the device's responsiveness in an event-based manner. In biological nervous systems, the Hebbian rule represents another mechanism for regulating synaptic weights based on temporal events[29,30]. If the presynaptic neuron repeatedly fires shortly before the postsynaptic neuron, the synaptic weight increases. Conversely, a reduction in synaptic weight occurs if the firing order is reversed. In our MIPM system, a programmable pressure series is designed to modify the conductance of memristor. Activation of the presynaptic neuron is modeled by a pressure signal in the stiff state, triggering negative spike train that enhances the memristor's conductance, leading to the long-term facilitation. Conversely, if the postsynaptic neuron is activated first, a pressure signal in the soft state triggers positive spikes that decrease the memristor's conductance, leading to the long-term depression.

## Reconfigurable tactile plasticity of MIPM
MIPM systems exhibit reconfigurable tactile plasticity owing to the programmable mechanogate and iontronic memristor. The mechanogate leverages the rearrangement of dynamic covalent bonds within the heterogeneous gel network to program the surface morphology[31], enabling the MIPM to achieve reconfigurable tactile perception. As

shown in Fig. 4a and Supplementary Fig. 18, the mechanogate successfully reconfigures the surface from flat to various surface structures, including microgrids, micropillars, and microtriangle through efficient dynamic transesterification and transamidation bonds exchange. Detailed experimental procedures are presented in the Methods and Supplementary Information. According to the results of finite element simulations, these different surface structures demonstrate distinct strain variations[32,33], which enable the mechanogate to sense pressure with reconfigurable sensitivities. Within the pressure range of 0–50 kPa, the sensitivity of positive piezoresistance increased from 5.28 to 11.92 kPa$^{-1}$. Concurrently, the sensitivity of negative piezoresistance ranges from −0.047 to −0.112 kPa$^{-1}$. The sensitivity variations introduced by programmable structures are correlated with the tactile sensing performance parameters of the memristors. Specifically, the MIPM with sensitive structure converts applied forces into high-frequency spike trains, leading to increased PSCs (Supplementary Fig. 19). Under identical applied pressure conditions, PSC generated by a memristor with a piezoresistive sensitivity of 11.92 kPa$^{-1}$ is 100% greater compared to that produced by memristors with a sensitivity of 5.28 kPa$^{-1}$ (Fig. 4b and Supplementary Fig. 20). Furthermore, the device's current decay process transitioned from volatile to non-volatile as sensitivity increased. These reconfigured surface structures exhibit excellent stability in piezoresistive characteristics, maintaining consistent performance at switching cycles between stiff and soft states (Supplementary Fig. 21). The corresponding neuromorphic signals also demonstrate remarkable reproducibility, with nearly the same PSC during repeated operations (Supplementary Fig. 22).

In the MIPM system, the memristors exhibit dynamically iontronic properties corresponding to multiple ion species, which can program the transconductance and the PSC of iontronic memristor. As shown in Supplementary Fig. 23, under identical ionic concentrations, the PSC exhibit significant variations with different ionic species. This is primarily attributed to the distinct coupling interactions between ions and PEDOT:PSS[34,35], which arise from differences in ion size and charge density. In our system, we further tune the tactile plasticity of memristors through programming ionic species and concentration. As demonstrated in Fig. 4c and Supplementary Figs. 24–26, with the increase in ionic species from a single ion ($Na^+$) to a multi-ion system including $Na^+$, $K^+$, $Ca^{2+}$, the coupling effect between ionic and electronic charges can be enhanced. This results in a considerable augmentation of PSC, reaching approximately 100% higher than that of the system containing only NaCl. Furthermore, the enhancement of the multi-ion hysteresis effect exerts additional influence on memristor plasticity, as evidenced by the augmented decay time of the PSC (Fig. 4d, e). This iontronic programmability enhances the reconfigurable plasticity of the MIPM. These reconfigurable features expand the capabilities and adaptability of tactile sensing devices in complex environments, contributing to research and development in tactile sensing and neuromorphic systems.

## MIPM system for a biohybrid perception-actuation circuit
In our case, the MIPM bridged to the interface of efferent nerves to construct a biohybrid perception-actuation circuit, which can perceive complex tactile information and precisely control various levels of limb movements through highly plastic neuromorphic signals (Fig. 5a). Specifically, the MIPM converts different forms of pressure signals into temporal neuromorphic signals with excitatory or inhibitory characteristics. These signals are then transmitted to the common peroneal and tibial nerves, which control the tibialis anterior and triceps surae muscles of the rat's hind limb[36], respectively, inducing antagonist muscle movements (Fig. 5b and Supplementary Fig. 27). As shown in Fig. 5c, neuromorphic signals with oscillation features can precisely modulate biological neural activity to control the downstream tissue motor. Neuromorphic signals with frequencies increasing from 6 Hz to 50 Hz, triggered by pressure stimulation rising from 2 kPa to 50 kPa in

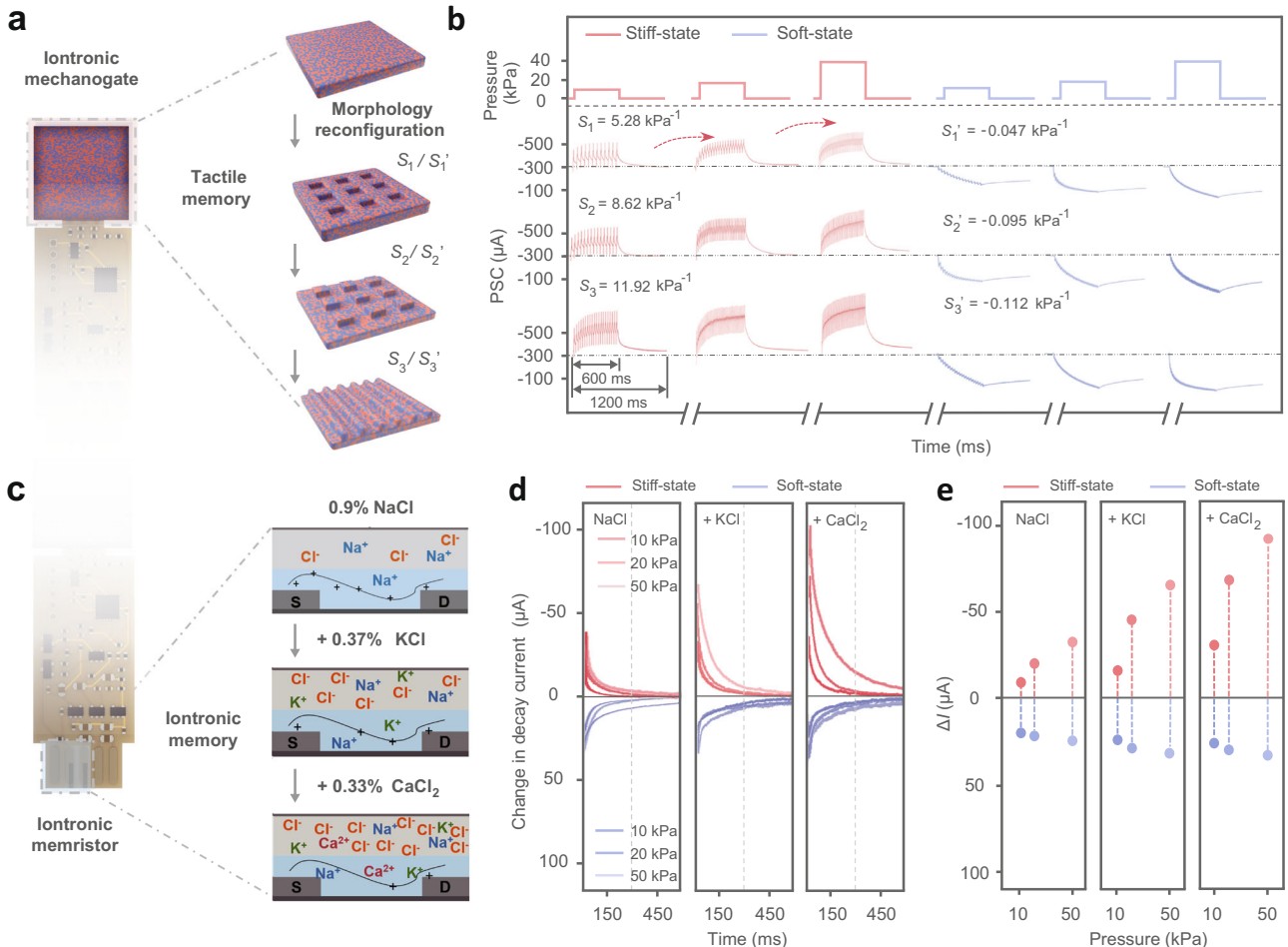

**Fig. 4 | Reconfigurable tactile plasticity of MIPMs. a** Structural illustration of the MIPM with the reconfigurable morphology iontronic mechanogate. The surface morphology can be programmed from flat to various structures, including micro-grids, micropillars, and microtriangle. **b** The postsynaptic current (PSC) signals based on the MIPM with different surface structures under pressure stimulations. **c** Structural illustration of MIPM with the reconfigurable iontronic memristor.

Through the control of multi-species ions as signal carriers, the MIPM exhibits programmable iontronic neuromorphics, resulting in multimode-reconfigurable processing of tactile signals. **d**, **e** The PSC decay curves and the absolute values of the change in PSC amplitude ($\Delta I$), respectively, demonstrating the influence of different ionic compositions on the MIPM plasticity.

stiff state, enhance the force output of activated muscle fibers by 500%. As a result, substantial changes in movement amplitude were observed, and the range of motion reached 47.5°. The integration of the MIPM with the rat's skeletal muscle system successfully translates the complex pressure information into tactile neuromorphic signals, inducing specific muscle movements. Different forms of tactile signals can be differentiated by the limb force signals of different muscle regions, in which flexion corresponds to excitatory neuromorphic signals to respond to the normal-temperature pressure stimulation, and extension is associated with inhibitory signals to respond to the high-temperature pressure stimulation (Supplementary Movie 1). Importantly, due to the cumulative effects of excitatory and inhibitory neuromorphic signals, the neuromorphic signals demonstrated temporal plasticity to response to complex pressure stimulations. Thus, the rat's movements exhibited a strong correlation with the historical activity (Fig. 5d). This plasticity enables the interfacing of dynamic tactile perceptions with diverse neuromodulations.

## Discussion

We have reported a MIPM system that incorporates programmable temporal-tactile neuromorphic plasticity. The system mimics biological mechanosensitive piezo channels and neuronal synapses, exhibiting high efficacy in translating complex tactile information into

neuromorphic signals. Our MIPM system captures and recognizes various pressure information through bidirectional piezoresistive signals of a switchable stiffness mechanogate. The integration of mechanogate with an oscillatory iontronic memristor allows the MIPM to demonstrate neuromorphic plasticity with excitatory and inhibitory properties. With its temporal-tactile neuromorphic plasticity, our system can successfully obtain complex tactile processing, memory and learning functions. Furthermore, it can bridge biological neural interfaces to construct a biohybrid perception-actuation circuit, achieving nuanced modulation of biological action behaviors based on temporal tactile plasticity. The MIPM system enables the recognition of complex tactile information, expanding potential applications in biomimetic mechanosensory systems, biohybrid robotics, tactile neuromorphic computing and smart wearable devices.

## Methods

### Fabrication of the bicontinuous heterogels

The bicontinuous heterogels were fabricated through an orthogonal polymerization-induced phase separation approach. Such as, 1 g of hexafluorobutyl acrylate (HFBA), 0.001 g phenylbis (2,4,6-tri-methylbenzoyl) phosphine oxide (PBPO, photoinitiator), 2 g of Poly(-caprolactone) diol (PCL-diol, Mn ~1000), and 1 g of IL were mixed and stirred until a uniform solution was achieved. Following this, 0.03 g of

tri-functional homopolymer of hexamethylene diisocyanate (THDI, Desmodur® N 3900) were promptly added. The bicontinuous hetero-gels were then formed through an orthogonal polymerization process,

involving UV light exposure at 405 nm and thermal treatment at 80 °C for 4 h. Subsequently, the cured bicontinuous heterogels were placed at 80 °C for 24 h.

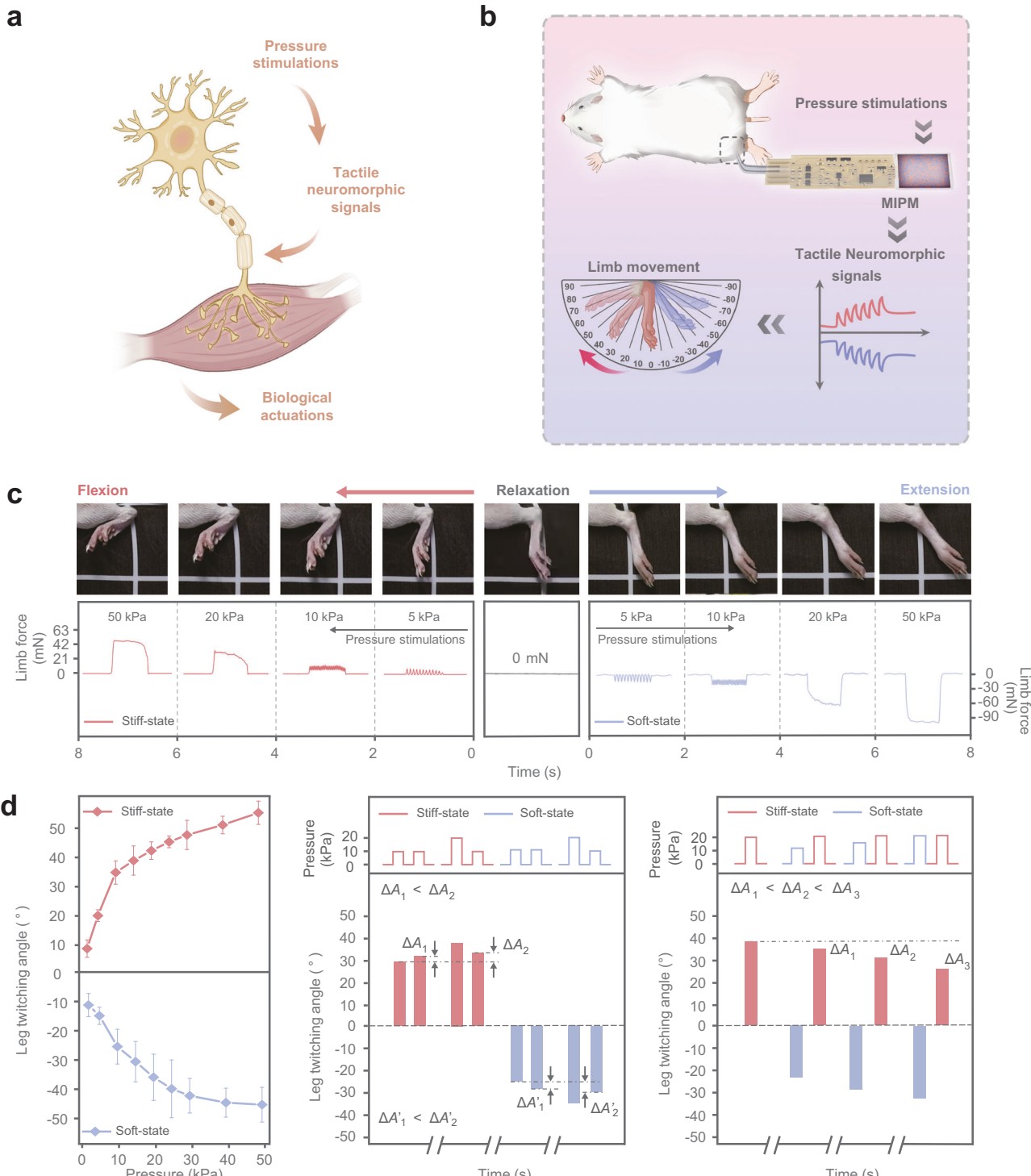

**Fig. 5 | MIPM system for a biohybrid perception-actuation circuit.**
**a**, **b** Schematic diagram of the biohybrid perception-actuation circuit. In this circuit, the MIPM acts as the interconnection system between tactility and biological neural interface. The neuromorphic signals with temporal plasticity are transmitted to neural interface, driving downstream tissue responses based on input tactile sti-mulations, thus facilitating various levels of limb movements. **c** Photos showing the various amplitudes of limb movement in response to different tactile neuro-morphic signals. The synchronized recorded muscle contraction reflects the

relationship between the different neuromorphic signals and various levels of limb movement. **d** Amplitude statistics of limb movement in response to temporal tactile neuromorphic signals correlated with different pressure inputs. Data points represent average values from five rats. Owing to the temporal plasticity, rats show limb movements dependent on historical activity (Δ𝐴, corresponding to the change in limb intrinsic twitching angle), achieving the adaptively tactile perception of dynamic pressure.

## Construction of oscillatory iontronic memristor

A micro-integrated pre-amplifier, including GPIOs, CPU, RCC, and PLL-equipped microcontroller, inverter, output amplifier, and iontronic memristor were assembled on a flexible polyimide substrate. The pre-amplifier, connected to a mechanogate, receives and amplifies bidirectional piezoresistive signal inputs to the microcontroller. These signals are then identified by GPIOs and encoded by the CPU, generating oscillatory pulses through the RCC. Depending on the polarity of the piezoresistive signal, the output oscillatory pulses are either inverted through a phase inverter to produce negative spike trains or directly transmitted to the output amplifier to generate positive spike trains. These positive and negative spike trains are further transmitted to downstream memristors to modulate the output of bipolarized neuromorphic signals. Detailed methods for the fabrication of the iontronic memristors are provided in Supplementary Information.

## Demonstration of the hybrid MIPM with biological systems

All animal experimental procedures used in the study were performed in compliance with animal welfare ethical regulations and approved by the Institutional Animal Care and Use Committee (IACUC) of Beijing Medconner (MDKN) Biotech Laboratory (MDKN-2023-055). Female Sprague Dawley rats were ordered from MDKN Laboratories with weight of 150 g at the time of arrival.

Before the surgery, the rat was deeply anesthetized with isoflurane (2 L min⁻¹ O₂ mixed with 3% isoflurane). For the sciatic nerve surgery, a 1.5 cm incision of the skin was made, and the sciatic nerve was exposed by separating muscles close to the femur. One end of the stimulating electrodes was connected to the MIPM, and the other end is linked to the tibial and the common peroneal nerve of the rat hindlimb. Corresponding to different pressure stimulations, the output neuromorphic signals of our MIPM system modulate the interface of the tibial and the common peroneal nerve, driving the movement of rat's hind limb. The forces generated by the leg muscles are monitored with BL-420 biological signal acquisition device in real time. This allowed us to correlate the pressure with the evoke leg twitching angle through MIPM and nerve stimulations.

## Reporting summary

Further information on research design is available in the Nature Portfolio Reporting Summary linked to this article.

## Data availability

The authors declare that all relevant data supporting the findings of this study are available in the paper and its Supplementary Information files or from the corresponding authors upon request. The raw data are available via Zenodo at https://doi.org/10.5281/zenodo.14575399. Source data are provided with this paper.

## Code availability

The code that supports the findings of this study is presented in the Supplementary Information. Source code is provided with this paper.

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

## Acknowledgements

The authors gratefully acknowledge Dr. Junchuan Yang and Meng Yuan from the Technical Institute of Physics and Chemistry, Chinese Academy of Sciences, for their valuable assistance in device fabrication and constructive suggestions during manuscript preparation. This work was supported by the National Natural Science Foundation of China (grant nos. T2425026 (Y.W.), 22275183 (Z.Z.), 52173190 (Y.W.), 22205077 (H.G.), 22102203 (Y.W.) and 21988102 (Y.W.)), the Ministry of Science and Technology of China (2018YFA0704803) (Y.W.), and Youth Innovation Promotion Association CAS (2018034) (Y.W.).

## Author contributions

Z.Z., Y.W. and M.L. conceived the idea of this research. X.W., Z.W. and H.G. co-designed the experiment. X.M., Z.W., H.L. and X.W. prepared the materials and devices, and performed the characterization measurements. Y.L. and W.D. carried out the theoretical simulation. H.S. and S.C. performed biological experiments. S.C. and Z.Q. analyzed and interpreted the results of biological experiments. H.G., Z.Z. and X.W. contributed to the experimental data analysis and mechanism investigation. X.W., Y.W. and Z.Z. wrote and revised the manuscript.

## Competing interests

The authors declare no competing interests.
