## [Transparent Peer Review file · Nature Communications]

Mechano-gated Iontronic Piezomemristor for Temporal-tactile Neuromorphic Plasticity

Corresponding Author: Professor Yuchen Wu

Version 0:

Reviewer comments:

Reviewer #1

(Remarks to the Author)

The present paper by Wei et al. presents mechano-gated iontronic piezomemristor (MIPM) with programmable temporal-tactile plasticity. The MIPM demonstrates bidirectional piezoresistive signals, resulting in wide-span dynamic tactile sensing. Excitingly, by bridging the bioneuron interface, the MIPM system possess the capacity to construct biohybrid perception-actuation circuit. While the results are interesting and potentially suitable for publication in a journal like Nature Communications. However, there are some critical issues that should be addressed before considering of publication.

1. Similar system studies have been carried out largely. Compared with the following studies (Nat. Biomed. Eng. 2023, 7, 511. Science 2018, 360, 998. Nat. Commun. 2020, 11, 1369. InfoMat 2023, e12429. Sci. Adv. 2019, 5, eaax4961. Sci. Adv. 2021, 7, eaabe3996. Science 2023, 380, 735. Nat. Commun. 2021, 12, 158. SmartMat, 2024, e1290.), what are the advantages and characteristics of the study? Secondly, for the gel, what's the difference between these studies (Sci. Adv. 2024, 10, eadl2737. Nat Commun. 2022, 13, 1743.)?
2. The authors propose the use of bicontinuous phase-transition heterogel as functional materials for piezoresistor. I recommend the authors to elaborate in the introduction section, why this material was chosen and what are the advantages of this material over other materials.
3. As shown in Fig. S3b, only 10 cycles of switching between rigid and flexible states of the MIPM are presented. Could you elaborate on the durability of the device? How does the repeated switching between the stiff and soft states affect the device? How does it impact lifespan of the device?
4. Largely gel sensors with high sensitivity have been developed (Nanoscale, 2021, 13, 19155. Carbon 2017, 113, 395e403. ACS Appl. Polym. Mater. 2022, 4, 3205–3216. Chem. Eng. J. 2024, 488, 150816.). In Fig. 2g, the illustration and Table S1 seems not correspond.
5. The MIPM system proposed in this paper can convert the bidirectional piezoresistive signals captured by the piezoresistor into positive and negative pulse trains, is this conversion process accomplished by the MCU? The amplitude and frequency of the positive and negative spike pulse trains formed during the conversion process are calculated based on what equation?
6. In Fig 4, the authors demonstrate the multimodal reconfigurable plasticity of MIPMs composed of mechanical gates with different microstructures and iontronic memristors with varying ionic doping. However, the significance and specific application directions of this multimodal reconfigurable plasticity were not mentioned, which is perplexing.
7. In Fig. S15, the author shows the changes of the gel with different shapes under the regulation of contact force. How to ensure the repeated stability of gel and the system? In addition, the author emphasizes that temperature affects the internal ion channels of gels, leading to different piezoresistive effects. In general, the voltage will affect the performance of ionic gels. It should be verified that no reactions (e.g. decomposition and REDOX) occur when a voltage is applied to the gel.
8. I suggest that the authors add a performance test of the memristor device to highlight the stability of the system, including typical I-V curves, endurance and stability tests of the memristor.

Reviewer #2

(Remarks to the Author)

This manuscript reports a bicontinuous phase-transition heterogel developed from vitrimer and ionic liquid as a switchable

stiffness-iontronic mechano-gate, to transduce pressure signal into bidirectional piezoresistive signals. By integrating the mechano-gated device with an oscillatory iontronic memristor, the system exhibited stiffness-induced bipolarized excitatory and inhibitory neuromorphic plasticity, thereby enabling the activation of temporal-tactile memory and learning functions. However, the material design and concept of the bicontinuous vitrimer heterogel for realizing temporal-tactile neuromorphic plasticity lack novelty because a similar concept has been reported recently (Sci. Adv. 10, ead12737 (2024)). Although the authors integrated this material with an iontronic memristor-based circuit, the memristor characterizations are arguable and the explanation of the system's mechanism is insufficient. In this regard, I conclude that this manuscript is not suitable for publication in high impact journals such as Nature Communications.

1. The explanation on the relationship between the material's structure and functionality in terms of bidirectional neuromorphic sensing is limited. Moreover, the importance or advantages of bidirectional neuromorphic sensing over monotonic neuromorphic sensing should be clearly discussed.

2. The authors should explain in detail the mechanism that causes the stiff-state condition to trigger negative spike train and the soft-state condition to trigger positive spike train enabling tunable neuromorphic excitatory/inhibitory PSC under increasing pressure, for better understanding of this system. Additionally, the explanations on the shortened ion channel and lateral expansion causing partial intercepting of the ion channel under strain should be supported with experimental data.

3. The authors discussed the demonstration of an iontronic memristor using PEDOT:PSS and ionic gel. However, the mechanism and characteristics of the iontronic memristor mentioned in the manuscript are insufficient. The authors need to provide more additional descriptions to clarify the following:

(i) In Fig. S10, the authors illustrated the structure of the iontronic memristor in the mechano-gated iontronic piezomemristor (MIPM). According to the proposed structure and the working mechanism of the PEDOT:PSS iontronic memristor discussed in the manuscript, the device is more identical to a PEDOT:PSS based OECT-synaptic device driven in the depletion mode than a memristor. Thus, the mechanism as well as the resulting data been attributed to memristor features are questionable. Therefore, the authors need to clarify the differences between existing OECT-synaptic devices and the inherent characteristics of this iontronic memristor.

(ii) In Fig. 4, the authors illustrated the impact of a multi-ion system on the iontronic memristor. However, the manuscript does not provide any details on the reason for the changes in PSC amplitude with respect to the different compositions. Additionally, the authors should include a comparative analysis using a single-ion species (e.g., ion gels containing only NaCl) at an equivalent ionic concentration. This comparison would help demonstrate that the observed effects are attributable to the multi-ion system rather than merely resulting from an increase in ion concentration. Also, explain how effective ion dissociation was achieved as there is a high probability of these different ions bonding to each other to form ionic clusters.

4. In Fig. 3, the authors demonstrated temporal-tactile plasticity based on the stiffness variation of the heterogel, as well as its learning functions. However, to implement the learning rules as shown in Fig. 3f, it is crucial for the developed heterogel to undergo a rapid phase change in response to temperature. It is generally known that the phase transition of conventional ion gels with temperature typically requires a certain amount of time. Therefore, the authors should provide a more detailed explanation of the learning characteristics in Fig. 3, supported by the physiochemical results and interpretations regarding the time-dependent phase transition of the heterogel.

5. The authors explained in Fig. 4a that "the mechanogate successfully reconfigures the surface from flat to various structures, including microgrids, micropillars, and microtriangle" by leveraging the rearrangement of dynamic covalent bonds within the heterogeneous gel network. However, there is no data or explanation on how these structures were obtained except the structural illustration in Fig. 4a. Please explain in detail how the mechanogate can reconfigure its surface morphology by answering the following questions:

(i) Were silicone molds utilized? What temperature conditions were used? How was the reversible reconfiguration to original state achieved?

(ii) Also, explain the reason for the variations in the tactile sensitivity of the programmed structures. Moreover, it is essential to use a unified pressure unit for consistency (e.g., kPa that accounts for actual contact area instead of force (N)).

6. The manuscript contains unexplained abbreviations (e.g. TPCL, etc.), typographical mistakes (e.g. inotropic, sercers, etc.), and grammatical errors, which may distract readers and hinder overall comprehension.

Reviewer #3

(Remarks to the Author)
Please see attachment

Reviewer #4

(Remarks to the Author)
I co-reviewed this manuscript with one of the reviewers who provided the listed reports. This is part of the Nature

Communications initiative to facilitate training in peer review and to provide appropriate recognition for Early Career Researchers who co-review manuscripts.

Version 1:

Reviewer comments:

Reviewer #1

(Remarks to the Author)

After carefully considering the peer review comments, the author has made the following revisions and improvements to the article:

- 1、 The author has supplemented a comparative analysis with existing literature, clearly articulating the innovation of this study, which is the use of bicontinuous heterogels as stiffness-controlled iontronic mechanical gates. This innovation endows the device with the capability of bidirectional stress perception, making it a notable highlight of this research.
 - 2、 In Supplementary Figure S6, the author presents cyclic experiments of stiffness regulation and supplements the electrical performance test data after multiple cycles. These results effectively confirm that the device maintains stable electrical performance after numerous stiffness transitions.
 - 3、 The author has elaborated on the process of signal encoding and modulation, allowing readers to gain a deeper understanding of the signal processing mechanism.
 - 4、 The author has added a discussion on the application prospects of MIPMs, highlighting how their characteristics significantly enhance the applicability and flexibility of tactile sensing devices in complex environments, opening up new research directions in the fields of advanced tactile sensing and neuromorphic systems.
 - 5、 The author has provided a detailed supplementary explanation of the synaptic plasticity mechanism of MIPMs, further enriching the theoretical depth of the paper.
 - 6、 In Supplementary Figure S11, the author provides the performance test results of the memristor, which helps to validate the stability and reliability of the device. Overall, the author has responded comprehensively and meticulously to the peer review comments, resulting in a significant improvement in the quality of the article.
- Therefore, I believe that this article fully meets the publication standards of Nature Communications and recommend that it be accepted.

Reviewer #2

(Remarks to the Author)

The authors have fully addressed our comments to the manuscript.
The revised version can be published in the Nature Communications journal.

Reviewer #3

(Remarks to the Author)

The authors have addressed most of the questions raised. However, the question regarding "how to precisely maintain these temperatures in practical applications" remains unanswered. Specifically, it is unclear whether temperature control is achieved through the human body's natural thermal regulation or by employing a wearable heater. Could the authors provide specific application scenarios where temperature regulation is necessary? Offering such examples would better highlight the novelty and significance of this method by demonstrating its practical or potential usefulness.

In Figure 3f, the authors used two heterogel mechanogates at different temperatures. However, in the demonstration, pressure was selectively applied to only one of the mechanogates. In a wearable application, both gates could potentially be subjected to pressure simultaneously, potentially negating the effect. The authors should address these considerations and explicitly state in the figure caption that two heterogel mechanogates were utilized. This clarification would enhance the figure's interpretability.

Reviewer #4

(Remarks to the Author)

Version 2:

Reviewer comments:

Reviewer #3

(Remarks to the Author)

The authors have fully addressed our comments to the manuscript.

Responses to Referees' Comments

Reviewer 1

The present paper by Wei et al. presents mechano-gated iontronic piezomemristor (MIPM) with programmable temporal-tactile plasticity. The MIPM demonstrates bidirectional piezoresistive signals, resulting in wide-span dynamic tactile sensing. Excitingly, by bridging the bioneuron interface, the MIPM system possess the capacity to construct biohybrid perception-actuation circuit. While the results are interesting and potentially suitable for publication in a journal like Nature Communications. However, there are some critical issues that should be addressed before considering of publication.

Response: We greatly appreciate reviewer's recognition of the novelty and potential impact of our work on mechano-gated iontronic piezomemristors (MIPMs) with programmable temporal-tactile plasticity. According to reviewer's constructive comments, we have carefully performed the related experiments and revised the manuscript. We believe that these revisions have significantly enhanced the quality and clarity of our work.

Comment 1: Similar system studies have been carried out largely. Compared with the following studies (Nat. Biomed. Eng. 2023, 7, 511. Science 2018, 360, 998. Nat. Commun. 2020, 11, 1369. Info Mat 2023, e12429. Sci. Adv. 2019, 5, eaax4961. Sci. Adv. 2021, 7, eabe3996. Science 2023, 380, 735. Nat. Commun.2021, 12, 158. Smart Mat, 2024, e1290.), what are the advantages and characteristics of the study? Secondly, for the gel, what's the difference between these studies (Sci. Adv. 2024, 10, eadl2737. Nat Commun. 2022, 13, 1743.)?

Response: Thanks for the reviewer's helpful comments. These referenced studies (Nat. Biomed. Eng. 2023, 7, 511. Science 2018, 360, 998; Nat. Commun. 2020, 11, 1369; Info. Mat. 2023, e12429; Sci. Adv. 2019, 5, eaax4961; Sci. Adv. 2021, 7, eabe3996; Science 2023, 380, 735; Nat. Commun. 2021, 12, 158; Smart. Mat, 2024, e1290.) primarily focus on external pressure-induced memristors where ion-gels are used as

external sensors or memristor gating components. As mentioned in our manuscript, we emphasize that existing tactile systems generally display monotonic neuromorphic features, as conventional gel-based sensors are limited to detecting unidirectional variations in stress intensity. By contrast, our research introduces a piezo-responsive memristor system designed to achieve bilingual temporal-tactile neuromorphics. This is accomplished using bicontinuous heterogels with bidirectional piezoresistive properties, functioning as distinctive mechanogating elements. Furthermore, the dynamic covalent bonding framework and iontronic mechanism of our system provide exceptional tunable reconfigurability of memristive behavior. This advancement considerably broadens the potential applications of tactile neuromorphic technology.

Regarding the differences in gel properties: a) Sci. Adv. 2024, 10, ead12737: This study primarily investigates the material perspective of bicontinuous heterogels with switchable stiffness. While we recognize the cited work (Sci. Adv. 2024, 10, ead12737.) as a significant contribution to gel materials science, our study introduces fundamental innovations in tactile-neuromorphic systems that extend well beyond gel material design alone. In our work, we redesign internal redesigns polymer structures and composites specifically to incorporate bicontinuous heterogels into mechano-gated iontronic piezomemristor systems, operating at biologically relevant temperatures as a crucial consideration for practical bioelectronic applications. b) Nat. Commun.2022, 13, 1743: This paper exhibits microphase-separated ion-gels with a shape-memory-enhanced unidirectional piezoresistivity. Differently, our work presents a novel bicontinuous phase structure capable of stiffness-induced bidirectional sensing, effectively advancing sensing capabilities. Importantly, temporal-tactile neuromorphic plasticity of the mechano-gated piezomemristor can also be achieved.

In our revised manuscript, we will further cite these references and provide related discussion of these studies in the introduction.

Comment 2: The authors propose the use of bicontinuous phase-transition heterogel as functional materials for piezoresistor. I recommend the authors to elaborate in the

introduction section, why this material was chosen and what are the advantages of this material over other materials.

Response: We thank the reviewer for these important comments. In our new introduction, we will further introduce the limitations of existing tactile sensors and recent developments in artificial tactile systems with neuromorphic features. According to the helpful suggestion of the reviewer, we have revised this section as follows:

“Conventional tactile sensors, particularly those based on conductive polymers, gel materials, and nanocomposites, transduce pressure signals into piezoresistive, capacitive, or piezoelectric outputs, which primarily reflect unidirectional pressure intensity variations⁷⁻¹³.”

“In response to varying tactile behaviors, these systems, with their limited sensing capabilities, struggle to provide elaborate temporal plasticity involving excitatory and inhibitory neuromorphic signals. This constraint hinders the development of complex tactile processing and memory functions, such as tactile memory erasure and reversible modulation.”

“This system utilizes a bicontinuous heterogel as a stiffness-governed iontronic mechanogate. In contrast to existing gel-based sensors that offer unidirectional tactile perception, the phase-transition-induced switchable stiffness of our heterogels introduces distinctive mechano-gated properties. These properties facilitate the transduction of complex pressure information into bidirectional (both negative and positive) piezoresistive signals within a biologically relevant temperature range of 25°C to 40°C (Fig. 1c).”

Comment 3: As shown in Fig. S3b, only 10 cycles of switching between rigid and flexible states of the MIPM are presented. Could you elaborate on the durability of the device? How does the repeated switching between the stiff and soft states affect the device? How does it impact lifespan of the device?

Response: We appreciate reviewer’s valuable comments. Due to the low permeability and low volatility of the ionic liquid components, the structure and composition of the heterogels remain stable over time. Additionally, the stiffness variation in the material

is governed by the phase transition characteristics of the polycaprolactone vitrimer composites. Since the phase transition is an intrinsic property of the polycaprolactone polymer network, this process exhibits inherent stability. As a result, the heterogel materials consistently maintain their phase-transition behavior and structural stability, ensuring reliable piezoresistive performance across various pressure conditions.

The Fig. R1 (new Fig. S5) demonstrates 100 cycles of switching between the stiff and soft states. We also conducted Differential Scanning Calorimetry (DSC) tests on the material, which showed that after 100 thermal cycles, the phase transition characteristics remained essentially unchanged. This consistency highlights the robustness and reliability of our material.

Fig. R1 (new Fig. S5) Stable phase-transition-induced switchable stiffness of the heterogel. **(a)** Storage moduli (G') of the heterogel at 100 continuous switching cycles between 25 °C and 40 °C. **(b)** Differential scanning calorimetry (DSC) curves comparing the 1st and 100th thermal cycles across the temperature range of 10-60 °C.

Moreover, the reliability of the bidirectional piezoresistive properties in our systems was thoroughly evaluated over 100 cycling processes, demonstrating excellent durability and a long operational lifespan (Fig. R2). Additionally, the dynamic covalent bonding present in the heterogels will offer extra self-healing properties, further extending their functional longevity.

To improve our manuscript, we offered the related discussion into the revised manuscript and added these experimental results Fig. R1 and Fig. R2 (new Fig. S5 and Fig. S6).

Fig. R2 (new Fig. S6) Reliable bidirectional piezoresistive properties of the MIPM's mechanogate during 100 continuous switching cycles between stiff state and soft state of bicontinuous heterogels.

Comment 4: Largely gel sensors with high sensitivity have been developed (Nanoscale, 2021, 13, 19155. Carbon 2017, 113, 395e403. ACS Appl. Polym. Mater. 2022, 4, 3205–3216. Chem. Eng. J. 2024, 488, 150816.). In Fig. 2g, the illustration and Table S1 seems not correspond.

Response: We appreciate this important comment of the reviewer. Regarding the high-sensitivity gel sensors mentioned, we acknowledge the significant contributions of these studies to the field of gel-based sensors. Therefore, we have added Fig. R3 (new Fig. 2g), Table R1 (new Table S1) and these references into the revised manuscript and Supplementary Information.

Fig. R3 (new Fig. 2g) The statistics of piezoresistivity features in existing typical piezoresistive materials and the heterogel. Distinguished from their unidirectional piezoresistive features, our mechanogate with the heterogel demonstrates a unique bidirectional stiffness-gated piezoresistivity.

Table R1. (new Table S1) The statistics of piezoresistivity features in existing typical piezoresistive materials and the heterogel.

	Positive sensitivity (kPa ⁻¹)	Negative sensitivity (kPa ⁻¹)	Reference
Liquid metal	4.31	/	1
	2.83	/	2
Ion hydrogel	0.31	/	3
	1.33	/	4
Nanocompsites	5.28	/	5
	/	-0.0054	6
IL gel	2.45	/	7
Conductive polymers	4.50	/	8
Aerogel	1.02	/	9
	0.31	/	10
Liquid metal	/	-0.025	11
Bicontinuous heterogels	4.34	-0.034	This work

The new references:

1. Yun, G. et al. Liquid metal hybrid composites with high-sensitivity and large dynamic range enabled by micro- and macrostructure engineering. *ACS Appl. Polym. Mater.* **3**, 5302-5315 (2021).
2. Han, X. et al. Green and stable piezoresistive pressure sensor based on lignin-silver hybrid nanoparticles/polyvinyl alcohol hydrogel. *Int. J. Biol. Macromol.* **176**, 78-86 (2021).
3. Shen, J. et al. A bioinspired porous-designed hydrogel@polyurethane sponge piezoresistive sensor for human-machine interfacing. *Nanoscale* **13**, 19155-19163 (2021).
4. Yang, H. et al. Coupling thermogalvanic and piezoresistive effects in a robust hydrogel for Deep-Learning-Assisted Self-Powered sign language and object recognition. *Chem. Eng. J.* **488**, 150816 (2024).
5. Park, S. et al. Piezo-impedance response of carbon nanotube/polydimethylsiloxane nanocomposites. *APL Mater.* **7**, 041118 (2019).
6. Liu, H. et al. Ambilateral convergent directional freeze casting meta-structured foams with a negative Poisson's ratio for high-performance piezoresistive sensors. *Chem. Eng. J.* **454**, 140436 (2023).
7. Mogli, G. et al. Ultrasensitive piezoresistive and piezocapacitive cellulose-based ionic hydrogels for wearable multifunctional sensing. *ACS Appl. Electron. Mater.* **5**, 205-215 (2023).
8. Teixeira, J. et al. Lanceros-méndez, Piezoresistive response of extruded polyaniline/(styrene-butadiene-styrene) polymer blends for force and deformation sensors. *Mater. Des.* **141**, 1-8 (2018).
9. Wang, Y. et al. Lightweight and elastic silver nanowire/PEDOT:PSS/polyimide aerogels for piezoresistive sensors. *ACS Appl. Polym. Mater.* **4**, 3205-3216 (2022).
10. Kang, W. et al. Photocrosslinked methacrylated carboxymethyl chitin hydrogels with tunable degradation and mechanical behavior. *Carbon* **113**, 395-403 (2017).
11. Yun, G. et al. Liquid metal composites with anisotropic and unconventional piezoconductivity. *Matter* **3**, 824-841 (2020).

Comment 5: The MIPM system proposed in this paper can convert the bidirectional piezoresistive signals captured by the piezoresistor into positive and negative pulse trains, is this conversion process accomplished by the MCU? The amplitude and frequency of the positive and negative spike pulse transformed during the conversion process are calculated based on what equation?

Response: In our MIPM system, the conversion of bidirectional piezoresistive signals into positive and negative pulse trains is accomplished by the microcontroller unit (MCU). As shown in Fig. R4 (new Fig. S7), the process involves several components: The pre-amplifier, connected to the mechanogate, receives and amplifies the bidirectional piezoresistive signal inputs. These amplified signals are then sent to the microcontroller, where they are continuously sampled through the ADC module and processed by the CPU. The conversion process relies on a differential threshold mechanism, where the system calculates the difference between consecutive ADC measurements to determine the signal direction and magnitude. When this difference exceeds a positive threshold, the system generates positive pulse trains, while differences below a negative threshold produce negative pulse trains. The microcontroller's RCC (Reset and Clock Control) generates oscillatory pulses based on the encoded signals. The amplitude of these spike pulses is fixed, with positive spike trains maintaining a constant amplitude of 0.4 V and negative pulse trains at -0.8 V. The frequency of the pulses is determined by a specific linear relationship with the ADC value, following the equation: $PWM_PRD = -29.2864 * AD_FINAL_VAULE + 113945.428$ or $PWM_PRD = 48.81855 * AD_FINAL_VAULE + 5000$. This relationship operates within carefully designed constraints, including a PWM frequency range of 8 Hz to 100 Hz and an analog input range of 0 V to 3 V. This comprehensive signal processing approach ensures that temporal and intensity information from the original piezoresistive signals is preserved through frequency modulation, while the amplitude is standardized to fixed values. The following code demonstrates the implementation and execution of this process. Next, these positive and negative spike trains are further transmitted to downstream memristors to modulate the output of bilingual neuromorphic signals.

The code is provided as follows:

```

/*****
 *
 * New Demand
 *
 *****/
If (TIM16_OUT_TRIG == 1) //ADC Final Count & generate PWM
{
    AD_FINAL_VAULE = AD_VAULE_ADDED / AD_COUNT;
    AD_COUNT = 0;
    AD_VAULE_ADDED = 0;

    /* AD RANGE 0 to +3V */
    /* AD VALVE RANGE 0-3720 */
    /* AD VALVE COMPARE -120nA to 0V */
    /* AD VALVE COMPARE 0nA to 1V */
    /* AD VALVE COMPARE 240nA to 3V */
    /* PWM Frequency RANGE 7.63Hz-100Hz */
    /* PWM_PRD Range 5000-65535 */
    If (AD_FINAL_VAULE >= 1653) //40nA
    {
        PMW_DIR = 1; //Pos State
        PWM_REGENERATE = 1;
    }
    else if (AD_FINAL_VAULE << 1240)
    {
        PMW_DIR = 0; //Neg State
        PWM_REGENERATE = 1;
    }
    TIM16_OUT_TRIG = 0;
}
If (PWM_REGENERATE == 1)
{
    HAL_TIM_Base_Stop_IT(&htim3);
    HAL_TIM_Base_DeInit(&htim3);
    HAL_GPIO_WritePin(GPIOB,GPIO_PIN_6,GPIO_PIN_RESET);
    PWM_Output_L_State = 0;
    HAL_GPIO_WritePin(GPIOB,GPIO_PIN_5,GPIO_PIN_RESET);
    PWM_Output_H_State = 0;
    If (PMW_DIR == 1)    PWM_PRD = -29.2864 * AD_FINAL_VAULE +
113945.428 ;
    If (PMW_DIR == 0)    PWM_PRD = 48.81855 * AD_FINAL_VAULE +
5000 ;
    MX_TIM3_Init();
    HAL_TIM_Base_Start_IT(&htim3);
    PWM_REGENERATE = 0;
}
AD_VAULE_LAST = AD_FINAL_VAULE;
}

```

Fig. R4 (new Fig. S7) Mechanism of signal conversion in the MIPM. A micro-integrated pre-amplifier, including GPIOs, CPU, RCC, and PLL-equipped microcontroller (MCU), inverter, output amplifier, and iontronic memristor were assembled on a flexible polyimide substrate. The pre-amplifier, connected to a mechanogate, receives and amplifies bidirectional piezoresistive signal inputs to the microcontroller, then generating the negative spike trains or positive spike trains. These positive and negative spike trains are further transmitted to downstream memristors to modulate the output of bilingual neuromorphic signals.

In the revised manuscript and Supplementary Information, these detailed explanations and the schematic (new Fig. S7) of this key aspect of our system have been included.

Comment 6: In Fig 4, the authors demonstrate the multimodal reconfigurable plasticity of MIPMs composed of mechanical gates with different microstructures and iontronic memristors with varying ionic doping. However, the significance and specific application directions of this multimodal reconfigurable plasticity were not mentioned, which is perplexing.

Response: We appreciate the insightful concerns raised by the reviewer. In our MIPM system, we integrated dynamic covalent bonding and iontronic features into tactile memristors, enabling the reconfigurable tactile plasticity.

This feature allows the device to dynamically adapt to various tactile inputs, offering versatile functionality for a wide range of practical applications. For instance, the reconfigurable plasticity enables the system to adjust its sensitivity based on

different stimuli, making it particularly valuable in applications where the adaptable tactile neuromorphic feedback is essential. By tuning the mechanogate surface microstructures, the device can be programmed to respond differently depending on the specific pressure it encounters, ensuring highly customized interactions with its environment. This is especially relevant for the development of artificial skin systems or neuromorphic devices that require real-time, adaptable sensory feedback. Additionally, optimizing this reconfigurability could further extend its applications to soft robotics. The reconfigurable plasticity ensures that the tactile memristor system can efficiently handle complex tasks, such as adaptive grip control and nuanced tactile recognition, significantly broadening its functionality and compatibility for advanced robotics and intelligent prosthetic systems. For our research, these reconfigurable features significantly expand the capabilities and adaptability of tactile sensing devices in complex environments. We believe that the related mechanisms and functions open new avenues for research and development in the field of advanced tactile sensing and neuromorphic systems.

According to the reviewer's suggestion, we have incorporated the significance and specific application directions of this reconfigurable tactile plasticity into the revised manuscript.

Comment 7: In Fig. S15, the author shows the changes of the gel with different shapes under the regulation of contact force. How to ensure the repeated stability of gel and the system? In addition, the author emphasizes that temperature affects the internal ion channels of gels, leading to different piezoresistive effects. In general, the voltage will affect the performance of ionic gels. It should be verified that no reactions (e.g. decomposition and REDOX) occur when a voltage is applied to the gel.

Response: Thanks for reviewer's insightful questions and comments. In the fields of shape memory materials, the dynamic covalent bonds acted as an efficient strategy for shape reconfiguration with stable stabilize the related structure (Sci. Adv. 2016, 2, e1501297; Angew. Chem. Int. Ed. 2016, 128, 11593-11597; Sci. Adv. 2018, 4, eaao3865.). In our system, the bicontinuous heterogel is composed of a vitrimer

network interpenetrated with an ion-liquid gel (ILgel) phase. The dynamic transesterification and transamidation bonds in the vitrimer network allow for surface microstructures reconfiguration with maintaining structural integrity (Fig. R5a and R5b). As shown in Fig. R5c (new Fig. S21), the heterogels with different programmed surface structures both exhibited their stable and efficient bidirectional piezoresistive property, which corresponded to the memristor results of our MIPM systems Fig. R6 (new Fig. S22). In the revised manuscript and Supplementary Information, we have added these results and related discussions to present the stability of heterogels and the system.

Fig. R5 (new Figs. S18a, b and S21) The reconfigurable surface microstructures of heterogels. (a) Schematic illustration of the dynamic covalent bonds in the vitrimer network. (b) Optical photographs of the heterogels with different surface microstructures. Scale bar, 100 μm . (c) Stable bidirectional piezoresistive property of these heterogels at switching cycles between stiff and soft states.

Fig. R6 (new Fig. S22) Reliable memristor results of the MIPM system with reconfigurable mechanogate at switching cycles between stiff and soft states.

Meanwhile, in our system, we have taken precautions to prevent any unwanted electrochemical reactions: a) Ionic liquid selection: We used an ionic liquid with a wide electrochemical window and high stability. b) Electrode selection and low operating voltage: The sensor systems utilize inert ITO electrodes to ensure no electrochemical reactions occur, and although the voltage can influence the gel's sensing performance, the operating voltage during the piezoresistive tests was carefully set to 1V to avoid any adverse interference. These comprehensive design considerations and control measures collectively ensure the long-term stability and reliability of our system during repeated operation.

Comment 8: I suggest that the authors add a performance test of the memristor device to highlight the stability of the system, including typical I-V curves, endurance and stability tests of the memristor.

Response: We thank reviewer's valuable suggestion regarding the performance testing of our memristor device. In response to your recommendation, we have added the two section results:

Heterogel stability: We have provided the DSC results with 100cycles phase transition (new Fig. S5). The cycles of bidirectional piezoresistive test are also added (new Fig. S6).

Memristor stability: We performed systematic electrical characterization through current-voltage (I-V) measurements over 100 continuous operation cycles. As shown in Fig R7 (new Fig. S11), the device exhibits highly reproducible I-V characteristics across a gate voltage range of ± 1.0 V. The consistent hysteresis loops demonstrate excellent switching stability of our memristor device. Furthermore, we performed multiple cycle testing to evaluate the performance stability of our system with different surface morphologies (new Fig. S22 and Fig. S23).

Fig. R7 (new Fig. S11) Current-voltage (I-V) curves of the memristor from the 10th to 100th cycle under ± 1.0 V gate voltage sweep.

These results and test details have added into our revised manuscript and Supplementary Information, providing a more comprehensive characterization of our work.

Reviewer 2:

This manuscript reports a bicontinuous phase-transition heterogel developed from vitrimer and ionic liquid as a switchable stiffness-iontronic mechano-gate, to transduce pressure signal into bidirectional piezoresistive signals. By integrating the mechano-gated device with an oscillatory iontronic memristor, the system exhibited stiffness-induced bipolarized excitatory and inhibitory neuromorphic plasticity, thereby enabling the activation of temporal-tactile memory and learning functions. However, the material design and concept of the bicontinuous vitrimer heterogel for realizing temporal-tactile neuromorphic plasticity lack novelty because a similar concept has been reported recently (Sci. Adv. 2024, 10, eadl2737). Although the authors integrated this material with an iontronic memristor-based circuit, the memristor characterizations are arguable and the explanation of the system's mechanism is insufficient. In this regard, I conclude that this manuscript is not suitable for publication in high impact journals such as Nature Communications.

Response: We appreciate the reviewer's valuable feedback on our manuscript. Based on his or her insightful suggestions, we have strengthened our manuscript to clarify and highlight the unique concepts and functionalities of the mechano-gated iontronic piezomemristors. While we recognize the cited work (Sci. Adv. 2024, 10, eadl2737.) as a significant contribution to gel materials science, our study introduces fundamental innovations in tactile-neuromorphic systems that extend well beyond gel material design alone. In our work, we constructed internal polymer structures to incorporate bicontinuous heterogels into mechano-gated iontronic piezomemristor systems, operating at biologically relevant temperatures as a crucial consideration for practical bioelectronic applications.

Meanwhile, we would like to clarify that the novelty of this manuscript lies in advancing beyond the known features of single-species gel materials and traditional memristor systems to propose a conceptually novel tactile neuromorphics by i) constructing bipolarized excitation and inhibition tactile-neuromorphic features that demonstrate high-order temporal learning and memory functions; ii) first introducing dynamic covalent bondings and iontronic mechanisms into tactile memristor systems,

which enables the reconfigurable tactile plasticity; and iii) developing a biohybrid perception-actuation system, which allows controlled modulation of biological neural interfaces. Thus, we expect that this work can represent a meaningful advancement in neuromorphic electronics and biomimetic tactile systems, serving as a foundation for further exploration in applications such as tactile neuromorphic computing, and biomimetic machines.

Comment 1: The explanation on the relationship between the material's structure and functionality in terms of bidirectional neuromorphic sensing is limited. Moreover, the importance or advantages of bidirectional neuromorphic sensing over monotonic neuromorphic sensing should be clearly discussed.

Response: We appreciate the reviewer's helpful comments. We would further clarify the relationship between the material's structure and functionality, particularly regarding bidirectional neuromorphic sensing and its advantages over monotonic neuromorphic sensing. The mechano-gated iontronic piezomemristor (MIPM) achieves bidirectional pressure sensing through a bicontinuous heterogel structure that integrates a soft ion-liquid gel (ILgel) with a stiffer vitrimer phase. This design enables a switchable stiffness mechanogate with distinct ion piezoresistive behaviors, stemming from the unique stress-deformation characteristics of ion transmission channel structures in its soft and stiff states. The resulting piezoresistive properties and three-dimensional finite element analysis clearly illustrate these switchable behaviors. The bidirectional piezoresistive signals that capture complex pressure information are transmitted to an oscillatory circuit, where they are converted into oscillatory spike trains- positive and negative signal spikes corresponding to the initial piezoresistive signal direction. The iontronic memristor further processes and modulates the postsynaptic current (PSC) based on the direction and intensity of the original piezoresistive signals. This transformation results in temporal-tactile neuromorphic plasticity, where piezoresistive signals are converted into bipolar oscillatory spike trains signals that facilitate complex neuromorphic signals, such as excitatory and inhibitory responses with temporal plasticity.

The structural framework of the MIPM is critical in producing both excitatory and inhibitory neuromorphic responses within a biologically relevant temperature range, constructing the complexities of natural tactile perception more effectively than traditional monotonic systems. The bidirectional neuromorphic sensing enhances the precision of tactile differentiation by supporting excitatory and inhibitory effects, offering a wider range of dynamic interactions compared to monotonic systems. The MIPM system can emulate bilingual short-term and long-term plasticity, facilitating high-order neuromorphic functions like BCM and Hebbian learning rules. As shown in Fig 1c, tactile signal conversion and transmission in the MIPM system was clearly reflected. Otherwise, for reported voltage-gated memristor systems with bidirectional neuromorphic, the bidirectional tactile neuromorphics induced by inherent mechanisms under external pressure field conditions is noteworthy and deserves attention. Meanwhile, on this basis, we also integrate dynamic covalent bonding and iontronic features into tactile memristors, enabling the reconfigurable tactile plasticity. The bidirectional neuromorphic sensing in our MIPM provides enhanced dynamic processing, adaptability, and a closer emulation of biological neural systems, underscoring the potential for complex tactile processing, adaptive learning applications, and bio-interfaced actuation control.

To enhance our article, we have added related discussion in the revised manuscript.

Comment 2: The authors should explain in detail the mechanism that causes the stiff-state condition to trigger negative spike train and the soft-state condition to trigger positive spike train enabling tunable neuromorphic excitatory/inhibitory PSC under increasing pressure, for better understanding of this system. Additionally, the explanations on the shortened ion channel and lateral expansion causing partial intercepting of the ion channel under strain should be supported with experimental data.

Response: We appreciate the reviewer's concern regarding the detailed mechanism of our MIPM system. In our system, the bidirectional piezoresistive signals originate from the unique stiffness-dependent synergistic effects within our bicontinuous heterogel structure. In the stiff state, the presence heteronetwork provided a contrasting

mechanical feature. When the vitrimer phase, acting as an ion shielding phase, was compressed axially, the deformation of the stiff vitrimer resembled folding, leading to limited lateral expansion of heterogel structures. The shortened ion transmission pathway within the ion-liquid gels induced a more pronounced change in ionic conductivity under applied stress, demonstrating the negative piezoresistivity behaviors. In the case of the softened state, the vitrimer phase became flexible and was easily deformed under compression, which affected the deformation experienced by the soft IL gel phase. Thus, the softening deformation partially intercepted the continuous ion-conducting pathways, further leading to positive piezoresistivity. These bidirectional piezoresistive signals are processed through an integrated oscillatory circuit, where they are converted into corresponding spike trains - negative spike trains in the stiff state and positive spike trains in the soft state. These spike trains further drive ionic modulation of PEDOT:PSS channels, transforming them into excitatory or inhibitory post-synaptic current (PSC) responses. The negative spike trains drive ion injection into PEDOT:PSS channels, leading to enhanced conductivity and resulting in excitatory PSC. Conversely, positive spike trains drive ion removal from PEDOT:PSS channels, resulting in decreased conductivity and inhibitory PSC.

According to the helpful suggestions from the reviewer, we will provide the new process diagram (Fig. R8) on these tactile signal conversion and transmission in the MIPM system. The pre-amplifier, connected to a mechanogate, receives and amplifies bidirectional piezoresistive signal inputs to the microcontroller, then generating the negative spike trains or positive spike trains. For the frequency response of these positive and negative spike trains, the implementation details, including the signal processing algorithms and execution code, are provided below to ensure reproducibility. These positive and negative spike trains are further transmitted to downstream memristors to modulate the output of bipolarized neuromorphic signals.

The code is provided as follows:

```

/*****
*
* New Demond
*
*****/

```

```

If (TIM16_OUT_TRIG == 1) //ADC Final Count & generate PWM
{
    AD_FINAL_VAULE = AD_VAULE_ADDED / AD_COUNT;
    AD_COUNT = 0;
    AD_VAULE_ADDED = 0;

    /* AD RANGE 0 to +3V */
    /* AD VALVE RANGE 0-3720 */
    /* AD VALVE COMPARE -120nA to 0V */
    /* AD VALVE COMPARE 0nA to 1V */
    /* AD VALVE COMPARE 240nA to 3V */
    /* PWM Frequency RANGE 7.63Hz-100Hz */
    /* PWM_PRD Range 5000-65535 */
    If (AD_FINAL_VAULE >= 1653) //40nA
    {
        PMW_DIR = 1; //Pos State
        PWM_REGENERATE = 1;
    }
    else if (AD_FINAL_VAULE << 1240)
    {
        PMW_DIR = 0; //Neg State
        PWM_REGENERATE = 1;
    }
    TIM16_OUT_TRIG = 0;
}
If (PWM_REGENERATE == 1)
{
    HAL_TIM_Base_Stop_IT(&htim3);
    HAL_TIM_Base_DeInit(&htim3);
    HAL_GPIO_WritePin(GPIOB,GPIO_PIN_6,GPIO_PIN_RESET);
    PWM_Output_L_State = 0;
    HAL_GPIO_WritePin(GPIOB,GPIO_PIN_5,GPIO_PIN_RESET);
    PWM_Output_H_State = 0;
    If (PMW_DIR == 1)    PWM_PRD = -29.2864 * AD_FINAL_VAULE +
113945.428 ;
    If (PMW_DIR == 0)    PWM_PRD = 48.81855 * AD_FINAL_VAULE +
5000 ;
    MX_TIM3_Init();
    HAL_TIM_Base_Start_IT(&htim3);
    PWM_REGENERATE = 0;
}
AD_VAULE_LAST = AD_FINAL_VAULE;
}

```

Fig. R8 (new Fig. S7) Mechanism of signal conversion in the MIPM. A micro-integrated pre-amplifier, including GPIOs, CPU, RCC, and PLL-equipped

microcontroller (MCU), inverter, output amplifier, and iontronic memristor were assembled on a flexible polyimide substrate. The pre-amplifier, connected to a mechanogate, receives and amplifies bidirectional piezoresistive signal inputs to the microcontroller, then generating the negative spike trains or positive spike trains. These positive and negative spike trains are further transmitted to downstream memristors to modulate the output of bilingual neuromorphic signals.

We acknowledge that direct in-situ observation of ion channel changes within gel materials at the nanoscale remains a significant experimental challenge due to current limitations in observational techniques for in-situ gel structures. However, our experimental piezoresistivity data clearly demonstrates the variations in ion transmission within the heterogels under different mechanical states. Additionally, to provide further evidence, we conducted a three-dimensional finite element analysis (FEA) of the deformation mechanics of the bicontinuous heterogel structure (Fig. R9). By simulating the structural response and incorporating key parameters such as density, elastic modulus, and Poisson's ratio for each phase, we constructed a robust finite element model on solid mechanics. Our simulation results indicate that in the stiff state, the high modulus of vitrimer phase restricts lateral deformation of IL-gel phase. Consequently, axial compression primarily shortens the ion transmission pathways, slightly increasing the effective cross-sectional area of the IL-gel phase with ion channels from 49.4% to 49.9% under applied pressure. This increase in cross-sectional area, coupled with shortened ion channels, is consistent with our experimental observations of negative piezoresistivity. Conversely, in the soft state, the significantly reduced modulus of vitrimer allows for substantial lateral expansion of heterogel materials under compression. This lateral deformation causes the ILgel phase to become partially interrupted and disconnected, resulting in a substantial decrease in the effective cross-sectional area from 49.4% to 31.2%. These simulation results align well with our observations of positive piezoresistivity and support the proposed mechanisms.

We will further add these explanations and the supporting data (new Fig. S7 and new Fig. 2c) in our revised manuscript to provide a clearer understanding of the related material structure and device functions.

Fig. R9 (new Fig. 2c) Three-dimensional finite element analysis of the bicontinuous heterogels illustrating the bidirectional stiffness-governed piezoresistivity of the MIPM's mechanogate.

Comment 3: The authors discussed the demonstration of an iontronic memristor using PEDOT:PSS and ionic gel. However, the mechanism and characteristics of the iontronic memristor mentioned in the manuscript are insufficient. The authors need to provide more additional descriptions to clarify the following:

(i) In Fig. S10, the authors illustrated the structure of the iontronic memristor in the mechano-gated iontronic piezomemristor (MIPM). According to the proposed structure and the working mechanism of the PEDOT:PSS iontronic memristor discussed in the manuscript, the device is more identical to a PEDOT:PSS based OECT-synaptic device driven in the depletion mode than a memristor. Thus, the mechanism as well as the resulting data been attributed to memristor features are questionable. Therefore, the authors need to clarify the differences between existing OECT-synaptic devices and the inherent characteristics of this iontronic memristor.

(ii) In Fig. 4, the authors illustrated the impact of a multi-ion system on the iontronic memristor. However, the manuscript does not provide any details on the reason for the changes in PSC amplitude with respect to the different compositions. Additionally, the

authors should include a comparative analysis using a single-ion species (e.g., ion gels containing only NaCl) at an equivalent ionic concentration. This comparison would help demonstrate that the observed effects are attributable to the multi-ion system rather than merely resulting from an increase in ion concentration. Also, explain how effective ion dissociation was achieved as there is a high probability of these different ions bonding to each other to form ionic clusters.

Response: We appreciate the reviewer's insightful comments. In neuromorphic electronics, OECT synaptic devices and memristors are not mutually exclusive concepts. OECT synaptic devices represent a specific implementation of artificial synapses using organic electrochemical transistors, where conductance can be modulated through electrochemical processes. A memristor is a device whose resistance depends on the history of current flow, encompassing various types of devices that exhibit memory effects in their resistance states (Chua, L. O. IEEE Trans. Circuit Theory 1971, 18, 507-519; Sangwan, V. K. et al. Nat. Nanotech. 2015, 10, 403-406.). When an OECT synaptic device demonstrates history-dependent resistance changes and memory effects, it can be legitimately characterized as a memristor. Until now, for the field of memristor devices, researchers focus on the fundamental design and related characteristic (memory and resistor) functions (Nat. Rev. Electr. Eng. 2024, 1, 286-299.).

We understand the reviewer's concern regarding the similarities to PEDOT:PSS-based organic electrochemical transistors (OECTs) driven in depletion mode. However, we would like to prefer to describe our device as a memristor, because its memory behavior is associated with front-end mechanogate that differentiate our device from conventional OECT-synaptic devices. They are primarily designed for sensing and rapid-response applications, focusing on transient conductance modulation. In contrast, our device is engineered to emulate tactile sensing behaviors in biological systems, capable of high-order temporal learning and memory functions (such as BCM and Hebbian learning rules). This capability sets our device apart as a true memristor rather than a simple OECT. Meanwhile, the dynamic covalent bond network and programmable iontronic features within the MIPM systems introduce additional

modulation pathways to realize reconfigurable plasticity and adaptive neuromorphic behavior, which surpasses the capabilities of standard OECT-based synaptic devices. Therefore, we chose the broader concept of memristor to describe our device. This classification aligns with the established theoretical framework of memristive systems while specifically highlighting its application in tactile sensing and memory operations.

To validate that the observed effects are indeed attributable to the multi-ion system rather than merely increased ionic concentration, we have conducted comparative studies using multi-ion species at equivalent ionic concentrations. As shown in the Fig. R10, when using our multi-ion system with the total ionic strength matching the original NaCl at concentrations, the PSC amplitude and temporal characteristics differ significantly from those observed in the NaCl system. The PSC amplitude is primarily governed by the effective ion transport within the PEDOT:PSS channel, which is influenced by both the ionic size and charge density of the constituent ions. In our system, K^+ ions, with their larger ionic radius, exhibit enhanced coupling effects compared to smaller cations like Na^+ , leading to more efficient modulation of the PEDOT:PSS channel conductivity. Additionally, the presence of Ca^{2+} ions, with their higher charge, exhibits enhanced capability to modulate the PEDOT:PSS channel conductivity, resulting in more pronounced PSC changes as evidenced by the larger ΔI_3 and $\Delta I'_3$ values observed in the multi-ion system. To address the reviewer's suggestion, we will include a related analysis in the revised manuscript. These results and test details have added into our revised manuscript and Supplementary Information (new Fig. S23), which will help demonstrate that the observed enhancements are indeed attributable to the synergistic effects of the multi-ion systems rather than simply resulting from a higher ion concentration.

Fig. R10 (new Fig. S23) Post-synaptic current (PSC) of the MIPM system measured in different ionic compositions with constant total ionic concentration at 50 kPa loading pressure.

To minimize the formation of ionic clusters between different ion species in our device, the ionic concentrations are deliberately maintained at physiologically relevant levels (approximately 0.01 M total ionic concentration). At these dilute concentrations, which are significantly below saturation limits, the formation of ionic clusters is thermodynamically unfavorable.

Comment 4: In Fig. 3, the authors demonstrated temporal-tactile plasticity based on the stiffness variation of the heterogel, as well as its learning functions. However, to implement the learning rules as shown in Fig. 3f, it is crucial for the developed heterogel to undergo a rapid phase change in response to temperature. It is generally known that the phase transition of conventional ion gels with temperature typically requires a certain amount of time. Therefore, the authors should provide a more detailed explanation of the learning characteristics in Fig. 3, supported by the physiochemical results and interpretations regarding the time-dependent phase transition of the heterogel.

Response: We appreciate the reviewer's important concern about the phase transition kinetics in relation to the learning characteristics. In our work, the MIPM exhibits stiffness-induced bipolar excitatory and inhibitory neuromorphic behavior. As shown

in Fig. 3a–3e, the in-situ phase-transition features of the heterogel mechanogate enable distinct PSC characteristics in both the stiff and soft states, corresponding to plasticity transitions. Fig. 3f demonstrates the MIPM system’s correlation plasticity effect under continuous dynamic tactile signals, allowing for controllable temporal-tactile memory functions. During this process, the formal uniformity of the time scale within transitions should be considered. It is true that the switching of stiffness states cannot be uniform due to heat conduction discrepancies, leading to variations in the time scale. To address this experimentally, we connected two heterogel mechanogates in different states in parallel, facilitating smooth, uniform transitions at consistent time intervals. Generally, the uniformity of time pulses is considered to influence a device’s memory effect. Our experimental setup mitigates time-based variability, offering clearer insights into the memory effect at uniform time intervals. Specifically, for the BMC and Hebbian learning rules, we applied pressure stimuli to soft/stiff-state mechanogates for identical durations, maintained consistent intervals, and then reapplied pressure to examine how the memory conductance states established by historical pressure stimuli influenced subsequent pressure responses. Further details on this process will be provided in the updated Supporting Information. Additionally, we would like to confirm that the memory retention time of our memristor device can extend to the minute level, significantly longer than the phase-transition time required for switching between stiffness states (Figs.R12). This retention time ensures that the MIPM completes its phase transition and heat conduction processes, supporting complex switching between soft and stiff states and the associated memory and learning effects. Next, to further enhance the precision and functionality of our system, we plan to design arrays of heterogel mechanogates and incorporate polymer systems with distinct phase transition characteristics. It aims to provide more refined temperature-dependent time control over the device’s behavior, thereby broadening its applicability in complex neuromorphic systems.

To enhance our article, we have added these results (new Figs. S4) in the revised manuscript.

Fig. R12 (new Fig. S4) Infrared thermography showing the heat conduction process of heterogel films with the thickness of 200 μm at 40°C, 50°C, and 60°C. The color mapping represents temperature change.

Comment 5: The authors explained in Fig. 4a that “the mechanogate successfully reconfigures the surface from flat to various structures, including microgrids, micropillars, and microtriangle” by leveraging the rearrangement of dynamic covalent bonds within the heterogeneous gel network. However, there is no data or explanation on how these structures were obtained except the structural illustration in Fig. 4a. Please explain in detail how the mechanogate can reconfigure its surface morphology by answering the following questions:

(i) Were silicone molds utilized? What temperature conditions were used? How was the reversible reconfiguration to original state achieved?

(ii) Also, explain the reason for the variations in the tactile sensitivity of the programmed structures. Moreover, it is essential to use a unified pressure unit for consistency (e.g., kPa that accounts for actual contact area instead of force (N)).

Response: We appreciate the reviewer’s detailed concerns regarding the surface reconfiguration process of the heterogel mechanogate. In our experiments, the solid

polytetrafluoroethylene (PTFE) molds with various surface patterns were employed as anti-structure imprinting molds. The dynamic covalent bond exchange processes, specifically transesterification and transcarbamoylation reactions within the vitrimer phase of the heterogel, were activated at 140°C. This dynamic covalent network enabled the material's shape reconfiguration, allowing us to achieve consecutive surface structure programming through cyclic hot pressing using PTFE molds at 140°C for 6 hours. To recover to the original smooth surface, we applied the same conditions using a smooth mold for hot pressing. According to the helpful suggestions of the reviewer, we will include the chemical mechanism and surface morphology measurements in Fig. R13 (new Figs. S18a, b), along with additional experimental details in revised Supplementary Information.

Fig. R13 (new Figs. S18a, b) The surface reconfiguration of heterogels. (a) Schematic illustration of the dynamic covalent bonds in the vitrimer network. (b) Schematic illustration of the surface reconfiguration of heterogels. (c) Optical photographs of the heterogels with different reconfigurable surface microstructures. Scale bar, 100 μm .

The different surface structures demonstrate distinct strain variations under the same pressure, which enable the reconfigurable mechanogate to sense pressure with different sensitivities. The different surface structures demonstrate distinct strain variations under the same pressure, which enable the mechanogate to sense pressure with reconfigurable sensitivities. because the contact area differed depending on the surface structure. Following your suggestion, we have consistently used pressure (kPa)

throughout our revised manuscript. The average pressure is defined as $P=F/A$, where F is the applied force and S is the planar area of the material under pressure.

Comment 6:

Response: Thanks for the reviewer's careful review and attention to detail regarding the manuscript's clarity and readability. We have thoroughly reviewed the entire manuscript to address all unexplained abbreviations, typographical mistakes, and grammatical errors. All abbreviations are now properly defined upon their first appearance in the text. For instance, "TPCL" has been spelled out as "three-phase contact line" at its first mention. We have also corrected typographical errors throughout the manuscript, including changing "inotropic" to "iontronic" and "sercers" to "serves." Additionally, we have carefully edited the manuscript for grammar and style to ensure clear scientific communication.

Response to Reviewer 3:

Wei et al. developed a mechano-gated iontronic piezomemristor that mimics the temporal plasticity observed in biological systems. By integrating iontronic mechanogate with an oscillatory memristor, the device mimics both excitatory and inhibitory neuronal activity. The main contribution of this study lies in the introduction of a stiffness-governed mechanical sensing material system with temperature-dependent piezoresistivity into an iontronic memristor, upon which excitatory or inhibitory neuromorphic behaviors can be realized separately. While the investigation is thorough, a major concern is the practical challenge of maintaining precise temperature control at 25°C and 40°C for reliable operation. The following specific comments must be addressed before the manuscript can be accepted.

Response: We appreciate your recognition of our study's main contribution in developing a mechano-gated iontronic piezomemristor (MIPM). We will address your concerns regarding temperature control in the following point-by-point response and provide additional context for our approach.

Comment 1: The experiments in the study are conducted at two discrete temperatures of 25°C and 40°C, respectively. As the response behaviors of the device are highly temperature-dependent, how to precisely maintain these temperatures in practical applications needs to be elaborated. We recommend that the authors also characterize the device responses at other temperatures. Additionally, when we use a finger to touch the hetero-gel, how does the skin temperature from the fingertip affect the device's performance?

Response: We appreciate this important comment from the reviewer. In our experiments, we precisely maintained discrete temperatures of 25°C and 40°C using temperature control equipment to demonstrate the concept and results of pressure-induced tactile neuromorphics at these temperatures. It should be noted that the operation of the MIPM system is not limited to these specific temperatures. Because in our systems, the bicontinuous heterogel sensors exhibited bidirectional piezoresistive function governed by their stiffness factor, which are driven by the phase-transition

properties of the vitrimer framework. As shown in the Figs. R14, both thermodynamic (DSC) and mechanical (rheological) tests indicate a broad operational temperature window for the soft and stiff states, ensuring functional stability over a range of temperatures in real applications.

Fig. R14 (new Fig. S3b) (a) Differential scanning calorimetry (DSC) thermogram of the heterogels on a temperature sweep in the range of 10 to 60 °C. (b) Storage moduli (G') of the heterogels on a temperature sweep in the range of 20 to 80 °C.

Meanwhile, we conducted piezoresistive tests at 10°C, 25°C, 30°C, 40°C and 60°C to further investigate the temperature effects on the stiffness-induced bidirectional mechanism, as suggested by the reviewer (Fig. R15). At 10°C, 25°C and 30°C, the material maintains its stiff state, exhibiting consistent positive piezoresistive response. In contrast, at 40°C and 60°C, the material transitions to its soft state, showing distinct negative piezoresistive behavior. The inset schematics illustrate the corresponding structural changes in the bicontinuous network. Furthermore, the device's response to temperature of 10°C, 25°C and 30°C remains in the stiff-state regime, ensuring consistent excitatory responses as demonstrated by the PSC characteristics under different pressure stimuli. When transitioning to the soft state at the temperatures of 40°C and 60°C, the device maintains stable inhibitory response. These results provide a comprehensive view of the MIPM's performance across a wider temperature spectrum.

We have added these experimental results (new Figs. S3b and S16) to the revised Supplementary Information and included a related description in the manuscript.

Fig. R15 (new Fig. S16) The piezoresistive characteristics of the MIPM system at 10°C, 25°C, 30°C, 40°C and 60°C.

Regarding the interaction with a fingertip, the skin temperature affects the device primarily through heat conduction, but the core interaction mechanism remains unchanged. At typical fingertip temperatures (around 30°C), the material stays in its stiffer state, resulting in negative piezoresistive signals. If the fingertip has been in contact with a warmer object, the transferred heat can raise the local temperature, and once it exceeds the phase-transition temperature threshold, the material switches to its soft state, resulting in positive piezoresistive signals. In our experiments, we used a

temperature-controlled pressure module to clearly demonstrate this effect. Next, to achieve directly control via hand temperature, we can refer to a study on low-temperature shape memory polymers (J. Polym. Sci. B Polym. Phys. 2016, 54, 1397-1404). By utilizing materials with a lower phase transition temperature, it would be possible to trigger the phase transition using body heat, specifically from hand temperature.

Comment 2: The main contribution of this study is to use temperature to alter the piezoresistive behaviors of the mechanogate, and thus to regulate the excitatory or inhibitory features of the memristor. Nevertheless, changing the temperature needs time, typically with slow responses. Another simpler approach can be directly changing the polarity of the driving voltage applied upon the mechanogate. For example, we can use positive voltage for excitatory and use negative voltage for inhibitory. What are the pros and cons of temperature-triggered state change versus simply changing the polarity of the driving voltage?

Response: Thanks for reviewer's very insightful and interesting comments regarding the distinction between the mechanogate approach in our system and voltage-driven memristors. We would like to further clarify our perspective.

We acknowledge that many memristor studies utilize voltage polarity to achieve excitatory or inhibitory features in neuromorphic systems. In these systems, the transitions are externally induced by voltage stimulation. However, the fundamental difference in our work lies in the fact that we utilize pressure-responsive stiffness modulation to spontaneously regulate excitatory and inhibitory neuromorphic features. This approach enables us to create temporal-tactile neuromorphic plasticity through the natural response of the material to external mechanical stimuli, rather than relying solely on external voltage input. Additionally, we are the first to integrate material chemistry mechanisms, specifically through dynamic covalent bonding and iontronic properties, to achieve tunable reconfigurability in plasticity. This represents a significant departure from conventional memristor systems, where excitatory and inhibitory features are often directly voltage-driven. While it is true that voltage polarity

changes allow for fast responses, and we acknowledge the advantages of rapid switching in voltage-gated systems, our system introduces a novel concept and mechanism by utilizing temperature-driven stiffness changes to mimic tactile neuromorphic behavior under pressure field stimulation. We hope that this work can offer a unique and innovative strategy for tactile neuromorphic systems. Moving forward, we will further optimize our systems by developing micro-array structures that can exhibit rapider response and more complex neuromorphic behaviors. By pre-programming specific excitation and inhibition points within array structures, we can achieve more advanced tactile functionalities. This could offer a broader range of applications beyond traditional responsive memristors, as we continue to explore more dynamic neuromorphic material and device systems.

Comment 3: How to alter the mechanical stiffness of the mechanogate instantly as desired, as the results in Fig. 3f and Fig. S13 presents instant shifting between excitatory and inhibitory behaviors? What is the time needed to interchange between these two statuses? How did you achieve the instant temperature switch in Fig 5d? The device response during temperature change needs to be studied.

Response: We appreciate the reviewer's important comments. In our work, the MIPM exhibits stiffness-induced bipolar excitatory and inhibitory neuromorphic behavior. As shown in Fig. 3a-3e, we demonstrate the MIPM's distinct PSC features in both the stiff and soft states, corresponding to related plasticity transitions. Fig. 3f demonstrates the correlation plasticity effect in the MIPM system under continuous dynamic tactile signals, thereby achieving controllable temporal-tactile memory functions. During this process, the formal uniformity of the time scale within transitions is also presented. It is true that the switching of stiffness states may not be perfectly uniform due to heat conduction discrepancies, which can lead to variations in the time scale. Experimentally, we connected two heterogel mechanogates with different states in parallel, allowing smooth and uniform transitions with the same time intervals. Further details about this process will be provided in the revised supplementary information. Meanwhile, we would like to clarify that the memory retention time of our memristor device can extend

to the minute scale, which is significantly longer than the time required for switching between stiffness state. This time buffer allows the MIPM to exhibit the complex switching between these two states and the associated memory and learning effects. In future developments, we can design the arrays structures of heterogel mechanogates that can offer more rapid and more precise control over this functionality. In Fig. 5d, the uniformity of time was not a primary consideration. Thus, the switching between different states in this case was achieved in situ.

Regarding the device's response to temperature changes, the phase-transition of the heterogel mechanogates is influenced by heat conduction and thermal diffusion, while our study primarily focuses on stiffness-state-induced tactile neuromorphic plasticity. According to the helpful suggestions, we will include infrared thermography data in the revised supplementary information, showing the rapid thermal response of heterogel films with the thickness of 200 μm at 40°C, 50°C, and 60°C (Fig. R17). Furthermore, our rheological data already clearly reflect the correlation between material stiffness and temperature responsiveness, providing a comprehensive understanding of the material's thermal mechanical behavior. We have added a description of the heterogel's stiffness switching characteristics in the revised Supplementary Information (new Fig. S4).

Fig. R17 (new Fig. S4) Infrared thermography showing the heat conduction process of heterogel films with the thickness of 200 μm at 40°C, 50°C, and 60°C. The color mapping represents temperature change.

Comment 4: In Fig. 4, the authors claim that the morphology and the ion species of the MIPMs are multimode and reconfigurable, which seems to be exaggerated. These words (multimode and reconfigurable) typically refer to reversible properties and behaviors in a single system. Since the authors are just investigating the effects of surficial structures and ion species on the device performance, it is not appropriate to describe such devices with multimode and reconfigurable. Maybe 'tunable, controllable, ... etc.' could be more precise. Also, the effects of morphology and ion species on the device performance are widely reported. Relevant references are suggested to be added accordingly.

Response: We understand your concern regarding the use of "multimode" and "reconfigurable" to describe the detailed functions of our MIPM systems. As noted by the reviewers, the words (multimode and reconfigurable) typically refer to reversible properties and behaviors in a single system. In our case, the functionalities are indeed achieved within one system. However, to avoid any potential overstatement, we will replace the terms of "multimode" in our manuscript.

In Fig. 4, we illustrate that the bicontinuous heterogel mechanogate can be programmed from a flat surface into various microstructures, such as microgrids, micropillars, and microtriangles. These reconfigurable surface structures are derived from the rearrangement of dynamic covalent bonds within the heterogeneous gel network, aligning with the characteristics of dynamic covalent bond shape-memory polymers (Sci. Adv. 2016, 2, e1501297; Angew. Chem. Int. Ed. 2016, 128, 11593-11597; Sci. Adv. 2018, 4, eaao3865). In our study, we demonstrated the integration of these reconfigurable surface features with the plasticity of the MIPM system. Additionally, we introduced various ionic species to modulate iontronic properties continuously. We anticipate that this integration of chemical mechanisms with

plasticity functionality will drive further advancements in the development of multifunctional memristor systems.

According to the reviewers' helpful suggestions about the effects of morphology and ion species on the device performance, we have added the following references:

32. Ruth, S. R. A. et al. Rational design of capacitive pressure sensors based on pyramidal microstructures for specialized monitoring of biosignals. *Adv. Funct. Mater.* **30**, 1903100 (2020).
33. Ruth, S. R. A. et al. Microengineering pressure sensor active layers for improved performance. *Adv. Funct. Mater.* **30**, 2003491 (2020).
34. Li, P. et al. Switching p-type to high-performance n-type organic electrochemical transistors via doped state engineering. *Nat. Commun.* **13**, 5970 (2022).
35. He, R. et al. Organic electrochemical transistor based on hydrophobic polymer tuned by ionic gels. *Angew. Chem. Int. Ed.* **62**, e202304549 (2023).

Comment 5: Minor points: "biologically compatible temperature range of 25 to 40°C," is misleading. This temperature range is outside normal human skin temperature or core body temperature range.

Response: Thank you for your valuable suggestion. We agree that the phrase "biologically compatible temperature range of 25 to 40°C" could be misleading. To improve accuracy, we will revise this to "a biologically relevant temperature range of 25 to 40°C".

Comment 6: Minor points: The mechanogate exhibits negative piezoresistivity at 25 °C and positive piezoresistivity at 40 °C. Thus, should there be a critical temperature point between 25 °C and 40 °C where the mechanogate shows zero piezoresistivity? What's this critical temperature?

Response: We appreciate the reviewer's insightful comment regarding the critical temperature point between 25°C and 40°C. This critical temperature corresponds to the phase-transition point, approximately 37.5°C for our material. To clarify, the stiffness at 25°C and 40°C characteristically represents two distinct mechanical states, as

confirmed by our DSC and rheological tests. At this phase-transition point, the piezoresistive characteristics become highly unstable. The piezoresistive behavior is determined by the deformation of ion channels within the ILgel phase under varying stiffness states. At critical phase-transition point, ion channel deformation during the piezoresistive process becomes unstable, resulting in both positive and negative piezoresistivity occurring simultaneously during the cycle. Despite the concept of zero piezoresistivity being intriguing, we mayn't use it to describe such behavior at the phase-transition point.

Comment 7: Minor points: The manuscript would benefit from further refinement to enhance clarity and readability.

Response: We appreciate the reviewer's valuable suggestions. The entire manuscript has been carefully reviewed to refine language, structure, and presentation, and we believe that these revisions will significantly enhance the readability and clarity of our work.

Comment 8: Minor points: Typos "severs as ion transport channels": This should be corrected to "serves as ion transport channels".

Response: Thank you for pointing out this spelling mistakes. The revised sentence is: "In these bicontinuous heterogels, the ILgel phase serves as ion transport channels and the vitrimer phase exhibits a switchable-stiffness feature that can influence ion transmission behavior in a piezoresistive process."

We are grateful for your meticulous review. We have carefully proofread the entire manuscript in our revision to ensure no similar typographical errors remain.

Responses to Referees' Comments

Reviewer 1:

After carefully considering the peer review comments, the author has made the following revisions and improvements to the article:

1. The author has supplemented a comparative analysis with existing literature, clearly articulating the innovation of this study, which is the use of bicontinuous heterogels as stiffness-controlled iontronic mechanical gates. This innovation endows the device with the capability of bidirectional stress perception, making it a notable highlight of this research.

2. In Supplementary Figure S6, the author presents cyclic experiments of stiffness regulation and supplements the electrical performance test data after multiple cycles. These results effectively confirm that the device maintains stable electrical performance after numerous stiffness transitions.

3. The author has elaborated on the process of signal encoding and modulation, allowing readers to gain a deeper understanding of the signal processing mechanism.

4. The author has added a discussion on the application prospects of MIPMs, highlighting how their characteristics significantly enhance the applicability and flexibility of tactile sensing devices in complex environments, opening up new research directions in the fields of advanced tactile sensing and neuromorphic systems.

5. The author has provided a detailed supplementary explanation of the synaptic plasticity mechanism of MIPMs, further enriching the theoretical depth of the paper.

6. In Supplementary Figure S11, the author provides the performance test results of the memristor, which helps to validate the stability and reliability of the device. Overall, the author has responded comprehensively and meticulously to the peer review comments, resulting in a significant improvement in the quality of the article.

Therefore, I believe that this article fully meets the publication standards of Nature Communications and recommend that it be accepted.

Response: We sincerely appreciate the reviewer for their previous insightful comments, which have greatly contributed to the improvement of our research work. Receiving

and addressing constructive criticism and suggestions for enhancement is invaluable in refining our study.

Reviewer 2:

The authors have fully addressed our comments to the manuscript.

The revised version can be published in the Nature Communications journal.

Response: We are deeply grateful to the reviewer for their insightful comments that significantly contributed to the advancement of our research work.

Reviewer 3:

The authors have addressed most of the questions raised. However, the question regarding “how to precisely maintain these temperatures in practical applications” remains unanswered. Specifically, it is unclear whether temperature control is achieved through the human body’s natural thermal regulation or by employing a wearable heater. Could the authors provide specific application scenarios where temperature regulation is necessary? Offering such examples would better highlight the novelty and significance of this method by demonstrating its practical or potential usefulness.

In Figure 3f, the authors used two heterogel mechanogates at different temperatures. However, in the demonstration, pressure was selectively applied to only one of the mechanogates. In a wearable application, both gates could potentially be subjected to pressure simultaneously, potentially negating the effect. The authors should address these considerations and explicitly state in the figure caption that two heterogel mechanogates were utilized. This clarification would enhance the figure’s interpretability.

Response: We are grateful to the reviewer for their important comments and appreciate the opportunity to clarify a crucial aspect of our work. The demonstration in our manuscript utilized specific temperatures to induce phase-transition mechanism of heterogels, thereby modulating the stiffness of the mechanogate and enabling advanced

higher-order temporal-tactile neuromorphic functionalities. However, this does not imply that practical applications of the device must rely on direct inherent temperature control. Except for Figure 3f, the MIPM device primarily senses the temperature of external objects during pressure interactions, rather than relying on the extra modulation of its own temperature.

A key innovation of our work is the implementation of phase-transition-induced stiffness gating to achieve temporal-tactile neuromorphic plasticity. This approach introduces a novel dimension to tactile devices by incorporating thermally induced mechanical transitions into neuromorphic systems, thereby eliminating the need for conventional electronic temperature sensors. The system can naturally sense the temperature characteristics of external objects (e.g., substance interfaces with varying temperatures), triggering stiffness changes that subsequently influence its tactile neuromorphic features.

For real applications, the device can act as a mechanosensory module, mimicking the dynamic adaptability of biological mechanoreceptors. By sensing the temperature of external objects, the material induces stiffness changes, enabling context-dependent neuromorphic responses. This capability can be integrated into biohybrid robotic systems to enhance tactile perception. On the other hand, inspired by tactile neural processing, the device introduces a new dimension for tactile neuromorphic computing. For example, our system could contribute to artificial dynamic neural networks by integrating temperature-sensitive temporal plasticity into neuromorphic computing architectures. The added thermal-mechanical coupling could enable novel computational paradigms (as outlined in related neuromorphic research, e.g., *Nature*, 2021, 589, 386-390). Furthermore, in wearable devices, the material's responsiveness to external stimuli—such as touch, pressure, or temperature—can be exploited for adaptive sensing and interaction. Shape memory and stiffness-changing materials have shown great potential for soft robotics, flexible interfaces, and wearable health monitoring devices (*Science*, 2020, 370, 961–965; *Sci. Adv.*, 2019, 5, eaaw1066). For instance, Prof. Bao et al. demonstrated simultaneous temperature and pressure sensing based on the ion relaxation dynamics of polymers. Similarly, our work leverages the inherent

thermodynamic properties of functional materials to ensure robust performance without relying on complex electronic sensing mechanisms. In new manuscript, we will further incorporate application perspectives to enhance the novelty and significance of this method, demonstrating its practical and potential usefulness across a range of scenarios. We have expanded the discussion section in our revised manuscript as “The MIPM system enables the recognition of complex tactile information, expanding potential applications in biomimetic mechanosensory systems, biohybrid robotics, tactile neuromorphic computing and smart wearable devices.”

In Figure 3f, the use of two heterogel mechanogates at different temperature states was intended to isolate the impact of temporal factors on memory and learning effects. The detailed reasons were discussed in previous response letter. Meanwhile, according to your suggestions, we have revised both the figure caption and manuscript to explicitly state this experimental configuration: Figure caption: “Temporal-tactile plasticity of MIPM with two heterogel mechanogates at different stiffness states demonstrating BCM learning rule and Hebbian learning rule.” Manuscript: “To eliminate the potential influence of temporal inhomogeneity switching on memory and learning effects, two heterogel mechanogates featured different stiffness states were used alternately within the MIPM system.” These revisions better clarify the purpose of the two mechanogate configuration and its selective activation in our experimental demonstration. We believe these modifications will enhance the figure’s interpretability and provide readers with a clearer understanding of the experimental design.

General Comment

Wei et al. developed a mechano-gated iontronic piezomemristor that mimics the temporal plasticity observed in biological systems. By integrating iontronic mechanogate with an oscillatory memristor, the device mimics both excitatory and inhibitory neuronal activity. The main contribution of this study lies in the introduction of a stiffness-governed mechanical sensing material system with temperature-dependent piezoresistivity into an iontronic memristor, upon which excitatory or inhibitory neuromorphic behaviors can be realized separately. While the investigation is thorough, a major concern is the practical challenge of maintaining precise temperature control at 25°C and 40°C for reliable operation. The following specific comments must be addressed before the manuscript can be accepted.

Specific Comments:

1. The experiments in the study are conducted at two discrete temperatures of 25°C and 40°C, respectively. As the response behaviors of the device are highly temperature-dependent, how to precisely maintain these temperatures in practical applications needs to be elaborated. We recommend that the authors also characterize the device responses at other temperatures. Additionally, when we use a finger to touch the hetero-gel, how does the skin temperature from the fingertip affect the device's performance?
2. The main contribution of this study is to use temperature to alter the piezoresistive behaviors of the mechanogate, and thus to regulate the excitatory or inhibitory features of the memristor. Nevertheless, changing the temperature needs time, typically with slow responses. Another simpler approach can be directly changing the polarity of the driving voltage applied upon the mechanogate. For example, we can use positive voltage for

excitatory and use negative voltage for inhibitory. What are the pros and cons of temperature-triggered state change versus simply changing the polarity of the driving voltage?

3. How to alter the mechanical stiffness of the mechanogate instantly as desired, as the results in Fig. 3f and Fig. S13 presents instant shifting between excitatory and inhibitory behaviors? What is the time needed to interchange between these two statuses? How did you achieve the instant temperature switch in Fig 5d? The device response during temperature change needs to be studied.
4. In Fig. 4, the authors claim that the morphology and the ion species of the MIPMs are multimode and reconfigurable, which seems to be exaggerated. These words (multimode and reconfigurable) typically refer to reversible properties and behaviors in a single system. Since the authors are just investigating the effects of surficial structures and ion species on the device performance, it is not appropriate to describe such devices with multimode and reconfigurable. Maybe 'tunable, controllable, ... etc.' could be more precise. Also, the effects of morphology and ion species on the device performance are widely reported. Relevant references are suggested to be added accordingly.

Minor points:

1. "biologically compatible temperature range of 25 to 40°C," is misleading. This temperature range is outside normal human skin temperature or core body temperature range.
2. The mechanogate exhibits negative piezoresistivity at 25°C and positive piezoresistivity at 40°C. Thus, should there be a critical temperature point between 25°C and 40°C where the mechanogate shows zero piezoresistivity? What's this critical temperature?
3. The manuscript would benefit from further refinement to enhance clarity and readability.

4. Typos "severs as ion transport channels": This should be corrected to "serves as ion transport channels".